# Molecular determinants of phase separation for *Drosophila* DNA replication licensing factors

**Matthew W Parker[1]\*[†], Jonchee A Kao[2][†], Alvin Huang[2][†], James M Berger[3], Michael R Botchan[2]**

[1]Department of Biophysics, University of Texas Southwestern Medical Center, Dallas, United States; [2]Department of Molecular and Cell Biology, University of California, Berkeley, Berkeley, United States; [3]Department of Biophysics and Biophysical Chemistry, Johns Hopkins School of Medicine, Baltimore, United States

**Abstract** Liquid–liquid phase separation (LLPS) of intrinsically disordered regions (IDRs) in proteins can drive the formation of membraneless compartments in cells. Phase-separated structures enrich for specific partner proteins and exclude others. Previously, we showed that the IDRs of metazoan DNA replication initiators drive DNA-dependent phase separation in vitro and chromosome binding in vivo, and that initiator condensates selectively recruit replication-specific partner proteins (Parker et al., 2019). How initiator IDRs facilitate LLPS and maintain compositional specificity is unknown. Here, using *Drosophila melanogaster* (*Dm*) Cdt1 as a model initiation factor, we show that phase separation results from a synergy between electrostatic DNA-bridging interactions and hydrophobic inter-IDR contacts. Both sets of interactions depend on sequence composition (but not sequence order), are resistant to 1,6-hexanediol, and do not depend on aromaticity. These findings demonstrate that distinct sets of interactions drive condensate formation and specificity across different phase-separating systems and advance efforts to predict IDR LLPS propensity and partner selection a priori.

**\*For correspondence:**
matthew.parker@
utsouthwestern.edu

[†]These authors contributed equally to this work

## Editor's evaluation

This paper studies the role of phase separation in replication initiation, with a focus on Cdt1. Sorting out the relative roles of phase separation and other mechanisms will require a detailed dissection of the amino acids driving phase separation, which can then be used to probe the role of phase separation in cells. Here, the authors perform extensive and comprehensive analyses of the amino-acid sequence requirements for Cdt1 phase separation finding important roles for charged and hydrophobic amino acids in mediating different aspects of the DNA-dependent phase separation observed.

## Introduction

The internal environment of a cell is compartmentalized both by membrane-bound organelles and by protein-rich and protein/nucleic acid-rich compartments that lack an enclosing membrane. These membraneless compartments are commonly called biomolecular condensates and form through an ability of their constituent factors to undergo liquid–liquid phase separation (LLPS) (reviewed in *Banani et al., 2017*). Cellular bodies that form by LLPS are often spherical due to surface tension minimization across the phase boundary (*Elbaum-Garfinkle, 2019*). Sphericity, however, is not a defining feature of biomolecular condensates and many variables can contribute to droplet (de)formation,

including the physical parameters of the liquid phase (e.g., surface tension, viscosity, and size) and external variables (e.g., application of a force and surface interactions). Nonspherical condensates often nucleate from or assemble along relatively rigid intracellular scaffolds. This effect is seen with cellular signaling complexes (*Banjade and Rosen, 2014*; *Li et al., 2012*; *Su et al., 2016*) and tight junctions (*Beutel et al., 2019*), which spread with the dimensions of the plasma membrane. Likewise, the chromatin-associated synaptonemal complex (*Rog et al., 2017*) and the perichromosomal layer (*Booth and Earnshaw, 2017*) are predicted to form a protein-rich phase that contours with the more rigid nature of their substrate, mitotic chromosomes (*Batty and Gerlich, 2019*; *Goloborodko et al., 2016*; *Houlard et al., 2015*; *Sun et al., 2018*).

We previously discovered that the factors responsible for initiating DNA replication in metazoans, in particular the fly homologs of the origin recognition complex (ORC) and Cdc6, as well as fly and human Cdt1, undergo DNA-dependent LLPS at physiological salt and protein concentrations in vitro (*Parker et al., 2019*). Recent studies have also demonstrated that human Orc1 and Cdc6 can phase separate in the presence of DNA (*Hossain et al., 2021*). In live cells, initiators first associate with chromatin in mitosis where they appear to uniformly coat, or 'wet', anaphase chromosomes (*Baldinger and Gossen, 2009*; *Kara et al., 2015*; *Parker et al., 2019*; *Sonneville et al., 2012*). These cytological observations are consistent with expectations for a protein that undergoes a condensation reaction with a relatively rigid intracellular scaffolding partner. The ability to condense on DNA is conferred by a metazoan-specific, N-terminal intrinsically disordered region (IDR) present in Orc1, Cdc6, and Cdt1. For Orc1, this region is required for binding chromatin in tissue culture cells and for viability in flies (*Parker et al., 2019*). In vitro assays show that the Orc1 IDR is dispensable for ATP-dependent DNA-binding by *Drosophila melanogaster* ORC and loading of the Mcm2–7 replicative helicase (*Schmidt and Bleichert, 2020*). However, at physiological ionic strength and protein concentrations, initiator IDRs drive initiator coalescence into a condensed phase that both supports ATP-dependent recruitment of Mcm2–7 and can be controlled by CDK-dependent phosphorylation (*Parker et al., 2019*), an event that directly regulates initiator mechanism (*Findeisen et al., 1999*; *Lee et al., 2012*). Together, the available data suggest that LLPS facilitates the formation of an enriched layer of initiation factors that coats mitotic chromosomes to enhance the kinetics of helicase loading, a process that is completed within a relatively narrow window of the eukaryotic cell cycle (*Dimitrova et al., 2002*; *Méndez and Stillman, 2000*; *Okuno et al., 2001*).

How metazoan initiator IDRs facilitate DNA-dependent LLPS at a molecular level is unknown. Protein multivalency underlies the formation of intermolecular interaction networks that can drive

**Table 1.** List of proteins that form liquid phase condensates sensitive to 1,6-hexanediol.

| Protein | Reference(s) |
|---|---|
| TDP43 | *Babinchak et al., 2019*; *Schmidt et al., 2019* |
| hnRNPA1 | *Molliex et al., 2015* |
| Tau | *Wegmann et al., 2018* |
| Huntingtin protein exon 1 | *Peskett et al., 2018* |
| Rnq1 | *Kroschwald et al., 2015* |
| BRD4 | *Sabari et al., 2018* |
| MED1 | *Cho et al., 2018*; *Sabari et al., 2018* |
| Hp1 | *Strom et al., 2017* |
| hnRNPA2 | *Lin et al., 2016* |
| TACC3 | *So et al., 2019* |
| FUS | *Kroschwald et al., 2017* |
| RPB1 | *Boehning et al., 2018* |
| Chromosome passenger complex | *Trivedi et al., 2019* |
| Synaptonemal complex | *Rog et al., 2017* |

LLPS. In many phase-separating systems, functional multivalency is contributed by the presence of one or more IDRs (*Li et al., 2012*). Although phase-separating IDR sequences vary between systems, some sequence features capable of supporting LLPS have begun to emerge. For example, many phase-separating IDRs have low amino acid sequence complexity (so-called low complexity domains [LCDs]) and are highly enriched for a few select amino acids (e.g., the nucleolar protein FIB1 [*Feric et al., 2016*], the P-granule protein Laf-1 [*Elbaum-Garfinkle et al., 2015*], the stress granule protein hnRNPA1 [*Molliex et al., 2015*], the transcription factor EWS [*Chong et al., 2018*], and the transcriptional coactivator MED1 [*Sabari et al., 2018*]). However, there exist important exceptions to this correlation, including phase separation by the intrinsically disordered nephrin intracellular domain (NICD) (*Pak et al., 2016*) and by metazoan replication initiators (*Parker et al., 2019*), both of which possess high sequence complexity IDRs. Thus, LCDs are but a subtype of phase-separating IDR.

Literature reports of phase-separating sequences are increasing rapidly, yet relatively few of these studies define the molecular basis for LLPS. An important goal of the field is to understand and ultimately predict the extent to which different disordered protein regions utilize a generally shared or individually distinctive set of cohesive interactions to drive phase separation. Interestingly, many condensates (*Table 1*) dissolve upon treatment with the aliphatic alcohol 1,6-hexanediol (1,6-HD), a compound that permeabilizes the nuclear pore by disrupting weak hydrophobic Phe-Gly ('FG')-repeat interactions that are prevalent in the assembly (*Patel et al., 2007*; *Shulga and Goldfarb, 2003*). The broad sensitivity of protein LLPS to 1,6-HD could be taken to indicate that phase separation mechanisms are generalizable, at least for certain IDR classes (*Kroschwald et al., 2017*). Consistently, IDR sequence aromaticity has been shown to be an essential feature in a variety of phase-separating systems (*Chiu et al., 2020*; *Chong et al., 2018*; *Lin et al., 2017*; *Nott et al., 2015*; *Pak et al., 2016*; *Qamar et al., 2018*; *Wang et al., 2018*). Nevertheless, other interaction types appear to predominate in other LLPS systems. For example, charged residues can facilitate phase separation either through homomeric protein–protein interactions (*Elbaum-Garfinkle et al., 2015*; *Mitrea et al., 2018*; *Nott et al., 2015*) or through heteromeric protein–protein (*Ferrolino et al., 2018*; *Pak et al., 2016*) and protein–nucleic acid interactions (*Lin et al., 2015*; *Zhang et al., 2017*). We note also that membrane-less organelles in vivo have a particular compositional bias that often underpins their utility, implying that there exists a molecular 'grammar' or 'sorting code' that accepts a particular subset of partner factors while excluding inappropriate factors. Consistent with this idea, we have observed that initiator IDRs can corecruit pathway-specific partner proteins but not other types of phase-separating sequences (such as human Fused in Sarcoma [FUS]) (*Parker et al., 2019*). An understanding of the molecular rules that govern IDR sorting mechanisms is still in its infancy.

Here, we use *D. melanogaster* Cdt1 as a model to dissect the molecular basis for DNA-dependent LLPS by metazoan replication initiator proteins. We find that phase separation by Cdt1 is unaffected by treatment with 1,6-HD and that sequence aromatics, which represent <3% of the total initiator IDR amino acids, are dispensable for condensate formation. We show that protein–DNA interactions are electrostatic in nature and contribute an adhesive force to LLPS, with DNA functioning as a counterion bridge. These interactions synergize with cohesive intermolecular IDR–IDR interactions, which are driven by the hydrophobic effect. Using mutagenesis, we demonstrate that intermolecular IDR–IDR interactions serve as the primary driving force behind initiator phase separation, with DNA-mediated bridging interactions playing a secondary but still facilitative role. Collectively, these studies provide a detailed picture of the biochemical mechanisms underlying IDR LLPS in replication initiation factors and reinforce the concept that biomolecular condensates can form by a variety of different molecular interaction types.

## Results

### Initiator LLPS is resistant to treatment with 1,6-HD and does not require aromatic residues

The IDRs of the metazoan replication initiation factors are necessary and sufficient for DNA-dependent LLPS (*Parker et al., 2019*). To understand the mechanisms that promote initiator condensation, we first set out to identify similarities in amino acid sequence composition between initiator IDRs and the IDRs of other condensate-forming proteins. We generated a sequence heatmap to compare the fractional representation of each amino acid in the IDRs of Orc1 (residues 187–549), Cdc6 (residues

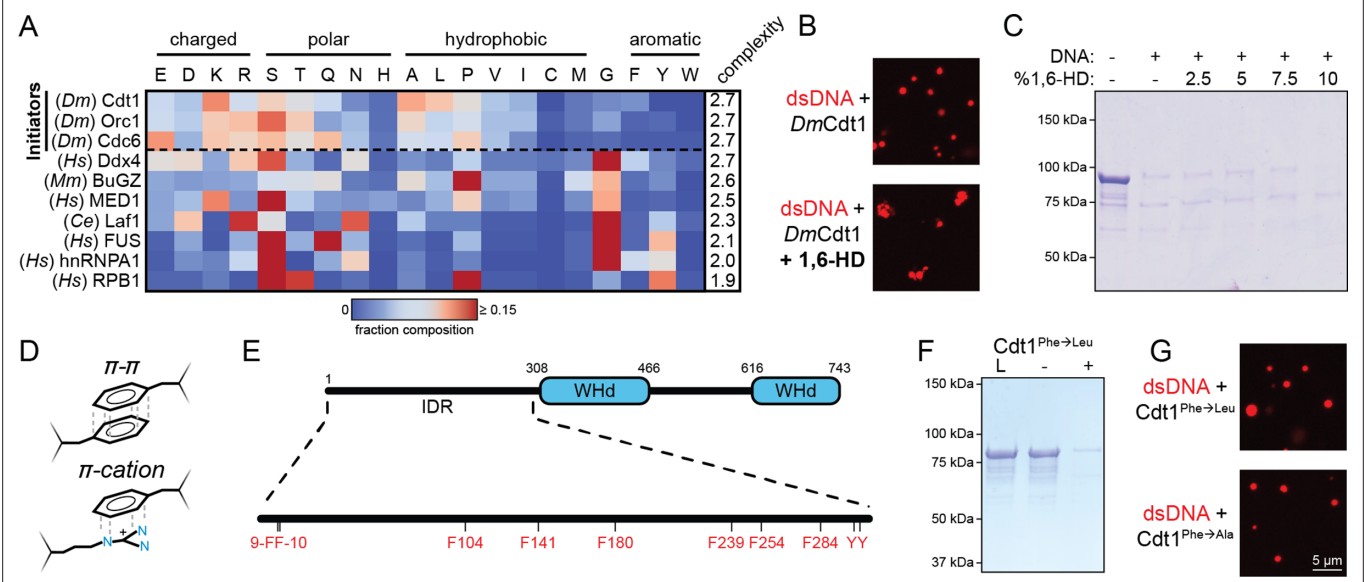

**Figure 1.** Initiator phase separation is insensitive to 1,6-hexanediol (1,6-HD) and does not require aromatic residues. (**A**) Heatmap representation of the amino acid composition of the three *Drosophila* initiation factors (Cdt1, Orc1, and Cdc6) compared to other phase-separating disordered sequences. (**B**) Confocal fluorescence microscopy images of condensates formed with 5 µM Cy5-double-stranded DNA (dsDNA) and 5 µM Cdt1 in the absence (top) and presence (bottom) of 1,6-hexanediol. (**C**) 2 µM Cdt1 was combined with 2 µM dsDNA and phase separation was assessed with a depletion assay in the presence of increasing concentrations (wt/vol) of 1,6-HD. Phase separation is indicated by protein loss. (**D**) Aromatic residues contribute to the phase separation of many condensate systems through their ability to participate in π–π and π–cation interactions. (**E**) Schematic of the *D. melanogaster* Cdt1 protein (top). Cdt1 contains two winged-helix domains (WHd, blue) and an N-terminal disordered domain (black line). The N-terminal intrinsically disordered region (IDR) contains multiple aromatic residues distributed throughout its length (bottom). (**F**) Depletion assay results assessing phase separation of the Cdt1 aromatic mutant, Cdt1$^{Phe→Leu}$. For this construct, all phenylalanine residues have been mutated to leucine. 'L' is a load control, '-' is in the absence of DNA, and '+' is in the presence of DNA. (**G**) Confocal fluorescence microscopy images of condensates formed with 5 µM Cy5-dsDNA and either Cdt1$^{Phe→Leu}$ (top) or Cdt1$^{Phe→Ala}$ (bottom). Gel and microscopy images are representative from three independent experiments.

The online version of this article includes the following figure supplement(s) for figure 1:

**Figure supplement 1.** Heatmap analysis and complexity calculations for phase-separating disordered domains.

**Figure supplement 2.** *D. melanogaster* origin recognition complex (ORC) and Cdc6 phase separation is resistant to treatment with 1,6-hexanediol.

1–246), and Cdt1 (residues 1–297) with the IDRs of a small suite of known condensate-forming proteins (human FUS [residues 1–293], Ddx4 [residues 1–260], hnRNPA1 [residues 178–372], DNA-directed RNA polymerase II subunit RPB1 [residues 1531–1970], and Mediator of RNA polymerase II transcription subunit 1 [MED1; residues 948–1574], *C. elegans* Laf-1 [residues 1–203], and mouse BuGZ [residues 63–495]) (**Figure 1A**). This analysis revealed major differences in the amino acid composition of initiator IDRs as compared to the other proteins analyzed, demonstrating that initiator IDRs represent an independent class of sequences. The most striking difference was the conspicuous absence of any one or more highly enriched residue within the initiator IDRs (dark red = fraction composition >15%). All other sequences in the comparison suite possess at least one highly enriched residue, most often serine, proline, and/or glycine. By contrast, initiator IDR composition is distributed more uniformly across all amino acids, apart from histidine, cysteine, and tryptophan (which are weakly represented among all IDRs assessed), and methionine, glycine, and aromatic residues (which are weakly represented in initiators but not necessarily other IDRs). Extending our heatmap analysis to a more diverse set of phase-separating disordered sequences (**Figure 1—figure supplement 1A**) revealed a similar trend, with most sequences being highly enriched in a small subset of amino acids. However, there are a minority of IDRs that show a more unbiased sequence composition similar to that of replication initiators, including the human NICD (**Pak et al., 2016**), TACC3 (**So et al., 2019**), and Tau (**Kanaan et al., 2020**).

The broad utilization of amino acid sequence space by initiator IDRs suggested that these sequences have higher complexity than many other phase-separating proteins. To quantitatively assess complexity, we calculated the informational entropy of each IDR sequence. In these

calculations, entropy represents the amount of information stored in a given linear sequence of amino acids as derived from the number of observed occurrences of each of the 20 amino acids, and has a theoretical range of 0–3 (*Wootton and Federhen, 1993*; *Figure 1—figure supplement 1B*). This analysis revealed that initiator IDRs have a higher sequence complexity (score = 2.7) than most other condensate-forming IDRs analyzed (14 of 17 total sequences). The human phase-separating proteins Ddx4 (score = 2.7) (*Figure 1A*), NICD (score = 2.8), TACC3 (score = 2.7), and Tau (score = 2.7) have similar complexity scores while still maintaining a unique sequence composition (*Figure 1—figure supplement 1A*). To contextualize these complexity calculations, we calculated the sequence complexity of all predicted disordered ($n$ = 3178) and ordered ($n$ = 10,884) sequences longer than 150 amino acids within the *Drosophila* proteome. Complexity scores for both classes of sequences adopted a negatively skewed distribution, with ordered sequences narrowly centered around 2.9 and disordered sequences more broadly distributed around a score of 2.7 (*Figure 1—figure supplement 1C*). These data reveal that phase-separating sequences show high variability in complexity and that initiator-type IDRs are among the most complex phase-separating sequences. The unique sequence features (i.e., composition and complexity) of initiator IDRs were a first indication that initiator LLPS might proceed through a mechanism distinct from many other model phase-separating proteins.

The aliphatic alcohol 1,6-HD can inhibit the self-assembly and phase separation of compositionally distinct classes of disordered sequences (see *Table 1* references). This behavior suggests that even for different IDR sequences, many use similar types of molecular interactions to drive LLPS. We therefore tested whether condensates formed from initiator IDRs are likewise sensitive to 1,6-HD. Initiator LLPS was directly visualized by fluorescence microscopy by individually mixing ORC, Cdc6, or Cdt1 with a Cy5-labeled 60 bp double-stranded DNA (Cy5-dsDNA). LLPS by all three proteins proved to be resistant to treatment with 10% 1,6-HD (*Figure 1B* and *Figure 1—figure supplement 2*), although 1,6-HD did lead to morphological changes in initiator/DNA condensates, including droplet clustering and, for ORC, a reduction in overall size. To confirm the microscopy results, we assessed the effect of 1,6-HD on Cdt1 partitioning into a condensed phase by a solution depletion assay. In this approach, LLPS is induced by the addition of dsDNA (60 bp), the denser phase-separated material is pelleted by centrifugation, and LLPS is assessed by protein depletion from the supernatant. Titrations of 1,6-HD from 0% to 10% (wt/vol) showed no ability to inhibit Cdt1 phase separation (*Figure 1C*), confirming that initiator phase separation is insensitive to this reagent.

Aromatic residues can play a critical role in driving protein LLPS, where they contribute multivalent π–π and π–cation interactions (*Figure 1D*; *Chiu et al., 2020*; *Chong et al., 2018*; *Lin et al., 2017*; *Nott et al., 2015*; *Pak et al., 2016*; *Qamar et al., 2018*; *Vernon et al., 2018*; *Wang et al., 2018*). However, the resistance of initiator LLPS to treatment with 1,6-HD (*Figure 1B, C*), as well as the low (<3%) aromatic residue content of initiator IDRs, suggested that these interactions may not be a primary driving force for phase separation. The IDR of Cdt1 contains only eight aromatic residues, all phenyl-alanine, that are relatively equally distributed throughout the region (*Figure 1E*). To test whether initiator LLPS is influenced by these aromatic residues, we mutated all phenylalanines in the Cdt1 IDR to leucine (Cdt1[Phe→Leu]) and assessed phase separation by both the depletion assay (*Figure 1F*) and fluorescence microscopy (*Figure 1G*, top panel). The replacement of Phe with Leu abolishes aromaticity in the IDR while maintaining a comparable level of hydrophobicity (phenylalanine hydrophobicity index = 2.8, leucine hydrophobicity index = 3.8, *Kyte and Doolittle, 1982*). Interestingly, the loss of aromaticity had no detectable effect on DNA-induced phase separation propensity. Mutating Cdt1's aromatic residues to alanine (Cdt1[Phe→Ala]) similarly did not block phase separation (*Figure 1G*, bottom panel). Together, these data show that initiator phase separation does not rely on aromatic residue-mediated interactions, consistent with its ability to resist treatment with 1,6-HD.

## Initiator IDR electrostatics promote coacervation with polyanionic scaffolds

The DNA dependency of initiator LLPS suggested that initiator droplets form by coacervation, a process in which two oppositely charged polymers drive the formation of a condensed phase (*Sing and Perry, 2020*). We therefore set out to test the sensitivity of Cdt1 LLPS to salt. Using the depletion assay, we measured phase separation at different concentrations of potassium glutamate (KGlu; 75, 150, and 300 mM) (*Figure 2A*). We observed Cdt1 LLPS at or below physiological levels of salt (150 mM) but a loss of condensation above this value. These data demonstrate that electrostatic

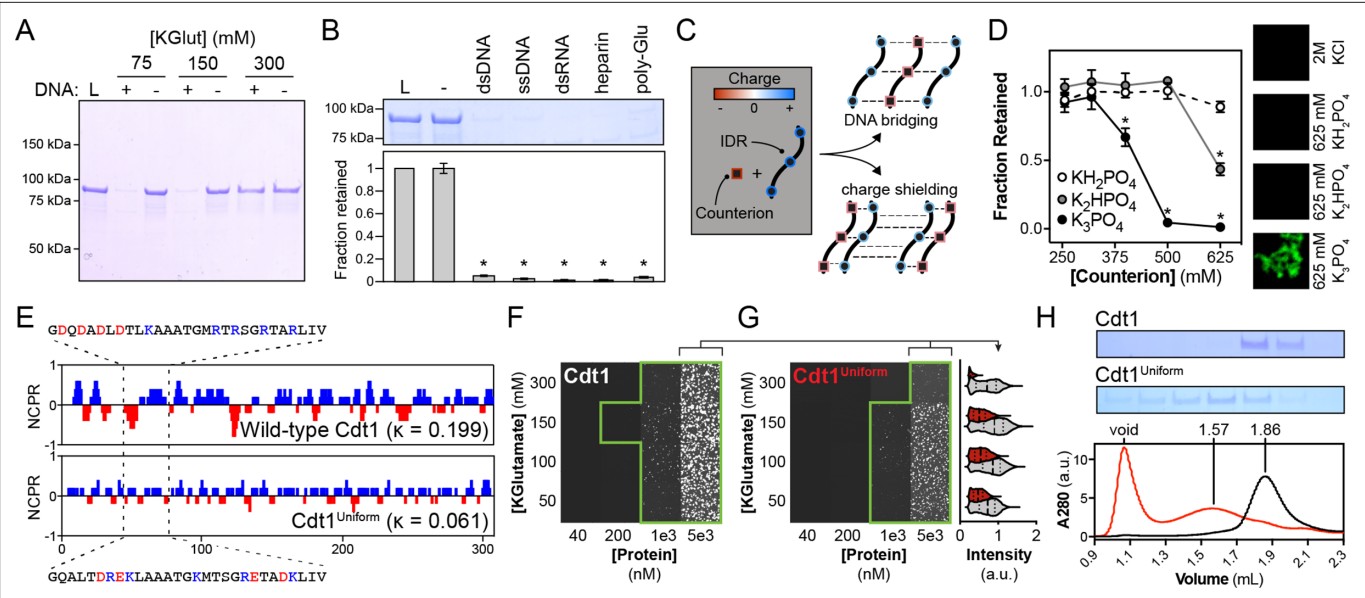

**Figure 2.** DNA contributes adhesive interactions to Cdt1 liquid–liquid phase separation (LLPS) by acting as a counterion bridge. (**A**) DNA-induced phase separation was assessed by the depletion assay in buffer containing 75, 150, or 300 mM potassium glutamate (KGlut). Cdt1 LLPS was observed at 75 and 150 mM KGlut, but not at 300 mM KGlut. (**B**) Multiple anionic polyelectrolytes were assessed for their ability to induce Cdt1 phase separation by the depletion assay. 'L' indicates a load control and '-' is in the absence of any anion polymer. Double-stranded DNA (dsDNA), single-stranded DNA (ssDNA), dsRNA, heparin, and poly-Glutamate (poly-Glu) all effectively induced Cdt1 LLPS. A $t$-test was used to calculate whether a significant change in pelleting resulted from the addition of each polyanion compared to the negative ('-') control. (**C**) Two models explaining the ability of DNA to induce Cdt1 phase separation. (Top) DNA (black/red) functions as a counterion bridge to facilitate inter-IDR (blue/red) interactions. (Bottom) DNA neutralizes inter-IDR electrostatic repulsion to drive self-assembly. (**D**) Monovalent ($KH_2PO_4$) and multivalent ($K_2HPO_4$ and $K_3PO_4$) phosphate counterions were assessed for the ability to induce Cdt1 phase separation by the depletion assay and fluorescence microscopy. For the line plot (left), the fraction of Cdt1 retained in the depletion assay was quantitated and plotted against counterion concentration (mM). For the microscopy experiment (right), 5 μM eGFP-Cdt1 was combined with 2 M KCl, 625 mM $KH_2PO_4$, 625 mM $K_2HPO_4$, or 625 mM $K_3PO_4$ (top to bottom) and assessed for droplet formation by confocal fluorescence microscopy. Concentrated and highly networked species were observed only in the presence of $K_3PO_4$. (**E**) Plot of the net charge per residue (NCPR) over a 5-residue sliding window for Cdt1 and the charged residue variant, Cdt1[Uniform]. (**F-G**) Phase diagrams for stoichiometric combinations of Cy5-dsDNA and Cdt1 (**F**) or Cdt1[Uniform] (**G**) at different protein and KGlutamate concentrations. Tiles bordered in green show conditions where phase separation was observed. Each tile is 75 × 75 μm. The mean signal intensity for Cy5-dsDNA within each droplet was quantitated for 5 μM Cdt1 (gray) and Cdt1[Uniform] (red) across the four salt concentrations. All pairwise comparisons are significantly different (p < 0.05 as determined by $t$-test). (**H**) Analytical size exclusion chromatography analysis of Cdt1 and Cdt1[Uniform]. Cdt1[Uniform] adopts a more extended conformation in solution as evidenced by a lower retention volume and copurifies with nucleic acid (260/280 = 0.56 and 0.79 for Cdt1 and Cdt1[Uniform], respectively). *p < 0.05, $t$-test. Gel and microscopy images are representative from three independent experiments.

The online version of this article includes the following figure supplement(s) for figure 2:

**Figure supplement 1.** Quantitative comparison of DNA enrichment in condensates formed by 5 μM wild-type and variant Cdt1s at 50, 100, 150, and 300 mM KGlutamate concentrations.

interactions are important for driving DNA-dependent phase separation by initiator proteins; however, it remained unclear whether specific Cdt1/DNA interactions, such as major- and minor-groove binding or ring-stacking interactions, might also be necessary for LLPS. To address this question, we assessed whether other anionic polyelectrolytes of distinct chemical composition could drive condensation. Double-stranded (ds) and single-stranded DNA (ssDNA), dsRNA, heparin, and poly-glutamate (each at 30 μg/ml) were combined with 2 μM Cdt1 and phase separation was assessed with the depletion assay (*Figure 2B*). All polyanions tested showed an equal propensity to induce Cdt1 phase separation, strongly supporting the notion that nonspecific electrostatic interactions are a significant driving force underlying DNA-induced initiator phase separation.

In principle, two mechanisms could account for the dependence of Cdt1 LLPS on the presence of a counterion (*Figure 2C*). First, a counterion may function as an intermolecular bridge to contribute adhesive forces that help promote initiator LLPS. Alternatively, electrostatic repulsion between the positive charges on the Cdt1 IDR might prevent the spontaneous formation of inter-IDR interactions and LLPS, and a counterion could help neutralize this repulsive force. To distinguish between these

mechanisms, we assayed phase separation by Cdt1 in the presence of mono- and multivalent counterions, with net charge ranging from −1 ($KH_2PO_4$) to −3 ($K_3PO_4$) (*Figure 2D*). We reasoned that if counterions simply reduce electrostatic repulsion, then the titration of a monovalent salt would be sufficient to shield the IDR's positive charge and drive phase separation. However, if counterions provide an intermolecular 'bridge' between basic residues within the Cdt1 IDR, then multivalent salts would be required to drive phase separation. Using the depletion assay we found that monobasic potassium phosphate (pH 7.5) was unable to induce phase separation of Cdt1 up to the highest concentration tested (625 mM). Conversely, we observed significant depletion of Cdt1 at the highest concentration of dibasic potassium phosphate (625 mM, pH 7.5), as well as for the three highest concentrations of tribasic potassium phosphate (400, 500, and 625 mM, pH 7.5). eGFP-tagged Cdt1 (eGFP-Cdt1) and fluorescence microscopy were used to confirm the depletion assay results. In the presence of either 625 mM mono- or dibasic potassium phosphate, eGFP-Cdt1 appeared monodisperse; however, tribasic potassium phosphate induced formation of a concentrated eGFP-Cdt1 species. Interestingly, this species was morphologically distinct from the round droplets of Cdt1 that are observed in the presence of DNA and appeared more

**Table 2.** Charged residue properties of metazoan and yeast initiator IDRs.

| Species | Protein | FCR | Kappa | pI |
|---|---|---|---|---|
| Fruit fly | Orc1 | 0.30 | 0.23 | 10.2 |
| | Cdc6 | 0.31 | 0.25 | 9.4 |
| | Cdt1 | 0.30 | 0.20 | 10.1 |
| Human | Orc1 | 0.31 | 0.21 | 10.6 |
| | Cdc6 | 0.25 | 0.21 | 10.6 |
| | Cdt1 | 0.28 | 0.26 | 10.6 |
| Frog | Orc1 | 0.32 | 0.29 | 9.7 |
| | Cdc6 | 0.25 | 0.21 | 10.8 |
| | Cdt1 | 0.30 | 0.20 | 10.1 |
| Zebrafish | Orc1 | 0.29 | 0.30 | 9.8 |
| | Cdc6 | 0.22 | 0.20 | 11.1 |
| | Cdt1 | 0.29 | 0.20 | 9.9 |
| Budding yeast | Orc1 | 0.48 | 0.43 | 4.7 |
| | Cdc6 | 0.32 | 0.59 | 6.3 |
| | Cdt1 (no IDR) | na | na | na |
| Fission yeast | Orc1 | 0.32 | 0.37 | 10.6 |
| | Cdc6 | 0.21 | 0.21 | 10.2 |
| | Cdt1 (no IDR) | na | na | na |

akin to aggregates than to droplets. These data indicate that the physiochemistry of a complexing polyanion can shape the properties of initiator condensates, consistent with previous observations for the complex coacervation of poly(proline-arginine) peptides (*Boeynaems et al., 2019*). Altogether, our findings strongly indicate that counterions contribute an adhesive force to initiator self-assembly, bridging between the IDRs of different Cdt1 protomers.

In addition to overall net charge, recent theoretical and experimental work has demonstrated that the patterning of charged residues within disordered domains can fine-tune phase separation propensity (*Lin et al., 2018*; *Nott et al., 2015*; *Pak et al., 2016*; *Paloni et al., 2020*). We assessed the distribution of charged residues within the Cdt1 IDR by calculating the net charge per residue (NCPR) over a 5-residue window (*Figure 2E*, top). Ionic residues within the Cdt1 IDR cluster into local regions with net-positive and net-negative charge. We predicted that if charge distribution were important for initiator phase separation, then charge patterning would be conserved across metazoan initiator IDRs. To test this idea, we performed a kappa value analysis of initiator IDRs. Kappa is a parameter between 0 and 1 that quantitatively describes the degree of mixing between oppositely charged residues within a sequence (well-mixed samples have a low kappa) and serves as a predictor of IDR conformational state (extended or collapsed) (*Das and Pappu, 2013*). Given the absence of sequence identity across metazoan initiator IDRs (Cdt1, Orc1, and Cdc6), we were surprised to find that the kappa values for metazoan initiator IDRs are similar (range 0.20–0.29), as are other physiochemical parameters (e.g., the overall fraction of charged residues, or FCR [range 0.22–0.32] and the isoelectric point, or pI [range 9.4–11.1]). None of these properties are preserved in budding yeast initiation factors (*Table 2*), which lack IDRs and do not undergo LLPS (*Parker et al., 2019*).

To directly test whether the distribution of charged residues within the Cdt1 IDR are important for its ability to form condensates, we produced a Cdt1 variant, Cdt1[Uniform], in which charged residues (K, R, D, and E) were uniformly distributed across the IDR sequence (kappa = 0.06), but where sequence composition and pI were maintained as per the wild-type protein. A plot of NCPR for Cdt1[Uniform]

demonstrates the loss of regions with high or low local net charge (*Figure 2E*, bottom). To quantitatively compare the phase separation propensity of Cdt1[Uniform] with that of wild-type Cdt1, we generated phase diagrams for these proteins, assaying the ability of Cy5-dsDNA to induce phase separation over a range of protein (40–5000 nM) and KGlutamate (50–300 mM) concentrations that span physiological conditions (in vivo, [Cdt1] ≈ 100 nM [*Parker et al., 2019*] and [salt] ≈ 150 mM [*Kretsinger et al., 2013*; *Park et al., 2016*]; *Figure 2F, G*). Wild-type Cdt1 phase separation was observed down to 200 nM protein at [KGlutamate] = 150 mM. Higher concentrations of Cdt1 resulted in phase separation at all concentrations of KGlutamate tested (*Figure 2F*). Surprisingly, Cdt1[Uniform] was still highly effective at inducing phase separation, although its critical concentration at [KGlutamate] = 150 mM was reduced to 1 µM and it was more sensitive to increasing concentrations of KGlutamate (*Figure 2G*). Quantitation of mean droplet intensity at 5 µM protein concentrations revealed a maximal Cy5-dsDNA signal at 150 mM KGlutamate for both proteins and an overall reduction in droplet intensity for Cdt1[Uniform] (*Figure 2G* and *Figure 2—figure supplement 1*). These results demonstrate that charge patterning in Cdt1 is not strictly required for phase separation, but that the charged residue distribution observed in the wild-type sequence favors lower LLPS critical concentrations and confers increased resistance to higher salt concentrations.

The dispensability of charged residue patterning for phase separation was initially surprising and prompted us to consider alternative roles for the observed native distribution of ionizable amino acids. A possible answer was gleamed from observations made during the purification of Cdt1[Uniform]. All proteins purified for this study were subject to a final polishing step over a size exclusion column, and it is at this step that we noticed a substantial difference in the elution profile of Cdt1 versus Cdt1[Uniform]. Wild-type Cdt1 eluted from an analytical sizing column at a volume consistent with its physical parameters (size and globularity) and had no contaminating nucleic acid (A260/A280 = 0.56, 100% protein by mass) (*Figure 2H*, black trace). Conversely, Cdt1[Uniform] eluted considerably earlier in a much broader peak, indicating that it possesses a more extended conformation compared to the native protein, and it coeluted with a small fraction of nucleic acid (A260/A280 = 0.79, or 98% protein by mass) (*Figure 2H*, red trace). Additionally, a significant fraction of the Cdt1[Uniform] protein eluted in the column's void volume, suggesting that this mutant shows higher aggregation propensity than wild-type Cdt1. Together, these results indicate that the distribution of charged residues within the Cdt1 IDR may encode some type of higher-order structural or conformational information that also regulates nucleic acid-binding capacity.

## Intermolecular IDR–IDR interactions can drive LLPS in the presence of crowding reagent

A parsimonious mechanism for LLPS by Cdt1 would rely exclusively on adhesive interactions (i.e., interactions between two dissimilar molecules) between the initiators and DNA (*Figure 3A*, top). However, it remained to be determined whether direct, inter-IDR contacts might provide an additional set of cohesive interactions (i.e., interactions between two similar molecules) that also promote LLPS (*Figure 3A*, bottom). To probe this issue, we asked whether Cdt1 could form condensates in the absence of DNA (or any other appropriate counterion) by the addition of a crowding reagent (*Figure 3B*). Using the depletion assay, we titrated PEG-3350 (from 0 to 12.5%, wt/vol, in 2.5% increments) against a fixed concentration of Cdt1 and assayed protein depletion from the supernatant. We observed a statistically significant reduction of Cdt1 from the supernatant starting at the lowest concentration of PEG-3350 (2.5%) that progressed as PEG concentrations were increased to a near-complete depletion of Cdt1 from the supernatant at the highest value tested (12.5% PEG-3350). To confirm that the PEG-induced loss of protein in the depletion assay represented phase separation (as opposed to precipitation), fluorescence microscopy was used to visualize mixtures of eGFP-Cdt1 and PEG-3350. Mixtures of Cdt1 and 4% PEG-3350 were sufficient to induce droplet formation (*Figure 3C*, bottom panel). Thus, in addition to poly-anions such as DNA, intermolecular Cdt1–Cdt1 interactions also provide a driving force for LLPS.

The ability of Cdt1 to self-associate prompted us to look for specific motifs within its IDR that could promote such interactions. As a first step, three different 100 amino acid segments were removed from the Cdt1 IDR (*Figure 3D*) and assessed for how the deletions impacted PEG-induced Cdt1 phase separation (*Figure 3E*). Deletion of IDR residues 1–100 (Cdt1[Δ1–100]) or 101–200 (Cdt1[Δ101–200]) had no discernible effect on Cdt1's propensity to phase separate in the presence of PEG. However, we

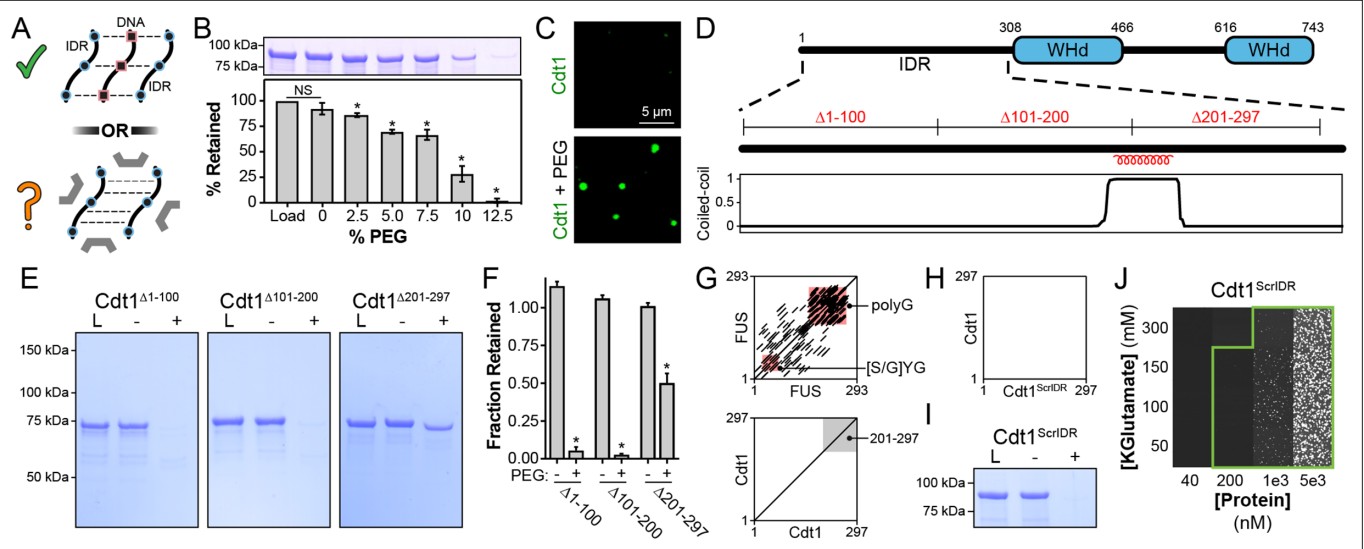

**Figure 3.** Inter-intrinsically disordered region (IDR) cohesive interactions drive Cdt1 phase separation in the absence of DNA. (**A**) (Top) DNA (black/red) can induce Cdt1 (black/blue) phase separation by acting as a counterion bridge. (Bottom) To what extent homotypic inter-IDR interactions affect phase separation was unknown. (**B**) Depletion assay to assess phase separation of 8 µM Cdt1 in the presence of increasing concentrations of PEG-3350 (0–12%, wt/vol). (**C**) Phase separation of 5 µM eGFP-labeled Cdt1 was assessed by confocal fluorescence microscopy in the absence (top) and presence (bottom) of 4% PEG-3350. (**D**) Schematic representation of the domain architecture of *D. melanogaster* Cdt1 (top) with coiled-coil domain prediction for the Cdt1 N-terminal IDR (bottom). The position of the three approximately 100 amino acid deletions used in (**E**) are indicated. (**E**) Cdt1$^{\Delta 1–100}$ (left), Cdt1$^{\Delta 101–200}$ (middle), and Cdt1$^{\Delta 201–297}$ (right) were assayed for the ability to undergo PEG-induced phase separation by the depletion assay. 'L' is a load control, '-' is in the absence of PEG, and '+' is in the presence of PEG. (**F**) Quantitation of depletion assay results presented in (**E**). A *t*-test was used to calculate whether a significant change in pelleting occurred in the presence compared to the absence of PEG. (**G**) Dotplot analysis (EMBOSS Dotmatcher; *Madeira et al., 2019*) of repetitive elements in the human Fused in Sarcoma (FUS) intrinsically disordered domain (top) and Cdt1 N-terminal IDR (bottom). The region representing Cdt1 amino acids 201–297 are highlighted with a gray background. The main diagonal represents sequence homology with itself and diagonals off the main diagonal represent repetitive motifs. Many repetitive motifs are apparent in FUS but are lacking in Cdt1. (**H**) Dotplot analysis of the Cdt1$^{ScrIDR}$, a Cdt1 variant with N-terminal IDR residues randomly scrambled, versus the wild-type Cdt1 IDR. (**I**) Depletion assay to assess phase separation of Cdt1$^{ScrIDR}$. 'L' is a load control, '-' is in the absence of PEG, and '+' is in the presence of PEG. (**J**) Phase diagram for stoichiometric amounts of Cdt1$^{ScrIDR}$ and Cy5-double-stranded DNA (dsDNA) at variable protein and KGlutamate concentrations. Tiles bordered in green show conditions where phase separation was observed. Each tile is 75 × 75 µm. *$p < 0.05$, *t*-test. Gel and microscopy images are representative from three independent experiments.

The online version of this article includes the following figure supplement(s) for figure 3:

**Figure supplement 1.** Depletion assay analysis of PEG-induced phase separation propensity for a Cdt1 variant lacking a predicted coiled-coil region.

**Figure supplement 2.** Dotplot analysis of various phase-separating intrinsically disordered regions (IDRs) to identify the presence of repetitive motifs.

**Figure supplement 3.** Cdt1$^{ScrIDR}$ condensates can recruit wild-type Cdt1.

observed a modest reduction in Cdt1 partitioning (~50%) into the condensed phase when residues 201–297 (Cdt1$^{\Delta 201–297}$) were deleted (***Figure 3F***). Although there is a predicted coiled-coil domain within the final third of the Cdt1 IDR (residues 196–223) that was suggestive of a possible self-association motif (***Figure 3D***, bottom), the deletion of this region alone had no impact on Cdt1's condensation properties (***Figure 3—figure supplement 1***).

Protein multivalency is a key property of proteins that undergo LLPS and frequently can arise from disordered sequences that contain short repetitive motifs (***Banani et al., 2017***). In the absence of obvious structural features to explain Cdt1 self-association, we attempted to identify repetitive motifs within the Cdt1 IDR – particularly within the 201–297 region – that might explain Cdt1 self-assembly. A prototypic example of repetitive IDR elements are those contained within human FUS, whose IDR contains multiple [S/G]YG motifs that are important for self-association (***Kato et al., 2012***; ***Luo et al., 2018***). These repeats, in addition to the protein's poly-glycine sequences, can be visualized diagrammatically by a Dotplot (***Madeira et al., 2019***; ***Figure 3G***, top). In this plot, a sequence is plotted against itself and a dot is placed at each point where a repetitive match of a specific length and threshold is found. We used this approach to search for repetitive sequence elements within the Cdt1 IDR. However, in stark contrast to FUS and many other condensate-forming proteins (***Figure 3—figure***

*supplement 2A-F*), we observed no repetitive motif(s) within the Cdt1 IDR (*Figure 3G*, bottom). This result led us to hypothesize that Cdt1 lacks repetitive sequences that would help promote self-association, and that it is the IDR amino acid composition alone that contains the necessary components to promote interprotomer interactions and phase separation. To test this idea, we synthesized and produced a Cdt1 variant, Cdt1^ScrIDR, where the order of the N-terminal IDR amino acids (residues 1–297) were randomly scrambled and then fused back to the wild-type C-terminal portion of the protein (residues 298–743). Dotplot analysis of the Cdt1 IDR versus that of the scrambled variant shows that no sequence motifs were inadvertently retained in Cdt1^ScrIDR (*Figure 3H*). The depletion assay was then used to ask whether this protein was still capable of phase separation. Surprisingly, Cdt1^ScrIDR showed robust phase separation propensity in the presence of a crowding agent, demonstrating that Cdt1^ScrIDR retains the self-associative capacity observed for wild-type Cdt1 (*Figure 3I*). We also assessed whether Cdt1^ScrIDR could undergo DNA-induced LLPS by assessing phase separation in the presence of Cy5-dsDNA over a range of protein and KGlutamate concentrations (*Figure 3J*). At [KGlutamate] = 150 mM, the Cdt1^ScrIDR critical concentration (200 nM) was equivalent to that of wild-type Cdt1 (*Figure 2F*) and, in fact, Cdt1^ScrIDR was able to phase separate over a broader set of experimental conditions. Notably, Cdt1^ScrIDR condensates were also able to recruit native eGFP-Cdt1, although the efficiency of recruitment was lower than that observed with wild-type Cdt1 condensates (*Figure 3—figure supplement 3A, B*). Collectively, these data demonstrate that it is the overall amino acid composition of initiator IDR sequences, not sequence order, that is essential for Cdt1's self-associative capacity and that composition alone can promote heteromeric inter-IDR interactions. However, the differences observed in the ability of eGFP-Cdt1 to be recruited into condensates formed by wild-type Cdt1 and Cdt1^ScrIDR also suggest that there exist specific sequence patterns that can fine-tune IDR association that have yet to be identified.

## Branched hydrophobics are disproportionately enriched in the Cdt1 IDR and drive LLPS

The Cdt1 IDR contains alternating blocks of net-positive and net-negative charge (*Figure 2E*). We reasoned that these interspersed blocks of opposite charge might facilitate inter-IDR interactions in the presence of a crowding reagent to drive phase separation; other condensate-forming proteins, such as *C. elegans* Laf1 (*Elbaum-Garfinkle et al., 2015*) and Ddx4 (*Nott et al., 2015*), have been shown to phase separate in such a manner. To test this idea, we compared the salt-sensitivity of PEG-induced Cdt1 LLPS to DNA-induced condensation (*Figure 4A*). At 75 and 150 mM KGlutamate, both DNA and PEG were comparable in their ability to induce Cdt1 phase separation. By contrast, salt concentrations above 150 mM KGlutamate abolished DNA-induced phase separation by Cdt1 but had no effect on PEG-induced LLPS. This result demonstrates that charge–charge interactions are not a major contributor to the inter-IDR interactions that drive PEG-induced phase separation. To test whether salt-insensitive inter-IDR interactions are a generally conserved feature of initiator phase separation, we assayed DNA and PEG-induced phase separation of both purified *Drosophila* ORC and Cdc6 at 150 and 300 mM KGlutamate concentrations (*Figure 4—figure supplement 1*). As seen for Cdt1, we observed that at 150 mM KGlutamate, either DNA or PEG are sufficient to induce Cdc6 and ORC phase separation, suggesting a role for both DNA-bridging interactions and direct inter-IDR interactions in metazoan initiator LLPS. Additionally, we found that DNA-induced phase separation is sensitive to 300 mM KGlutamate while PEG-induced phase separation largely is not. These data support a generally conserved mechanism of phase separation for metazoan Cdt1, Cdc6, and ORC.

We next set out to test whether initiator inter-IDR interactions might be driven by the hydrophobic effect. We first asked whether initiator IDRs are uniquely enriched in hydrophobic residues. However, when we compared the hydropathy of initiator IDRs to all ordered and disordered segments longer than 100 amino acids within the *D. melanogaster* proteome (14,002 ordered domains and 4957 disordered domains), we found that initiator IDRs were closer to the hydropathy of the median IDR (average of −1.04 for Orc1, Cdc6, and Cdt1) than to either the lower (−1.24) or upper (−0.80) quartile of the data (*Figure 4B*). This result led us to take a more unbiased approach and compare initiator IDR sequence composition (fraction average of each amino acid across Orc1, Cdc6, and Cdt1 IDRs) to the average composition of all ordered and disordered domains longer than 100 amino acids (*Figure 4C*). We found that initiator IDRs are more highly enriched in both the three most hydrophobic residues (isoleucine, valine, and leucine) and the three least hydrophobic residues (glutamate,

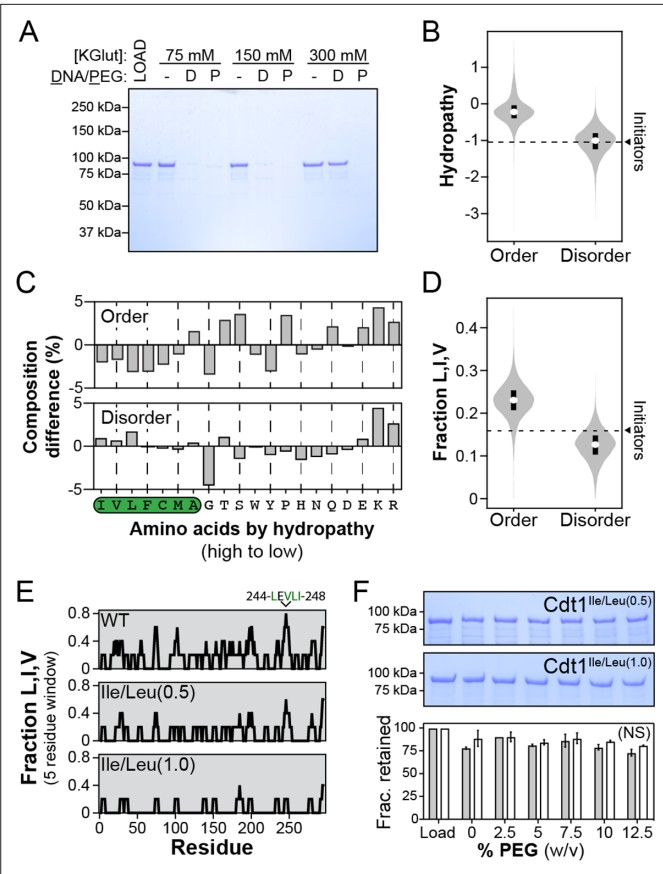

**Figure 4.** Branched hydrophobic residues underlie Cdt1 homotypic inter-intrinsically disordered region (IDR) interactions. (**A**) The depletion assay was used to assess the salt-sensitivity of PEG-induced phase separation side-by-side with DNA-induced phase separation. 'LOAD' is a load control, '-' is in the absence of DNA and PEG, 'D' indicates the addition of DNA, and 'P' indicates the addition of PEG-3350. Cdt1 phase separation was assessed at 75, 150, and 300 mM KGlutamate ('KGlut'). (**B**) Comparison of initiator IDR hydropathy (dashed line indicates the average hydropathy of the Orc1, Cdc6, and Cdt1 IDRs) to the hydropathy of predicted ordered domains ('Order') and disordered domains ('Disorder') proteome wide. (**C**) Percent difference in initiator IDR amino acid composition relative to all predicted ordered (top) and disordered (bottom) domains proteome wide. The amino acids are ordered from high to low hydropathy (left to right) and those with a positive hydropathy value are indicated by a green background. (**D**) Comparison of initiator IDR fraction Leu, Ile, and Val (dashed line indicates the average fraction-L,I,V of the Orc1, Cdc6, and Cdt1 IDRs) to the fraction-L,I,V of all predicted ordered domains ('Order') and disordered domains ('Disorder') proteome wide. (**E**) To assess the distribution of branched hydrophobic residues, the fraction-L,I,V was calculated over a 5-residue sliding window for the Cdt1 IDR (top). Two mutants were made with either half (Cdt1$^{Ile/Leu(0.5)}$) or all (Cdt1$^{Ile/Leu(1.0)}$) of Leu and Ile residues mutated to Ala, and the distribution of L,I,V was calculated for each (middle and bottom, respectively). (**F**) The depletion assay was used to assess the phase separation capacity of Cdt1$^{Ile/Leu(0.5)}$ (top gel) and Cdt1$^{Ile/Leu(1.0)}$ (bottom gel) in the presence of increasing concentrations of PEG-3350 (0–12%, wt/vol). The fraction of protein retained at each concentration of PEG-3350 was quantitated. A *t*-test was used to calculate whether the different PEG concentrations resulted in a significant change in pelleting relative to the no PEG control. No significant ('NS') dose-dependent trend was identified for the effect of PEG on Cdt1$^{Ile/Leu(0.5)}$ (grey bars) or Cdt1$^{Ile/Leu(1.0)}$ (white bars) phase separation. Gel images are representative from three independent experiments.

The online version of this article includes the following figure supplement(s) for figure 4:

**Figure supplement 1.** DNA-induced phase separation of origin recognition complex (ORC) and Cdc6 is sensitive to salt but PEG-induced liquid–liquid phase separation (LLPS) is not.

lysine, and arginine) compared to other disordered domains (*Figure 4C*, bottom). We also compared the total fraction of Leu, Ile, and Val (fraction-L,I,V) residues in the initiators to all ordered and disordered domains proteome wide. We found that the initiator fraction-L,I,V was within the top 25% of all disordered domains (*Figure 4D*). By comparison, initiator IDRs are relatively deficient in glycine and, to a lesser extent, uncharged polar residues, compared to other disordered sequences (*Figure 4C*, bottom).

The higher fraction of branched hydrophobic residues within initiator IDRs prompted us to test whether these amino acid types promote cohesive inter-IDR interactions that drive Cdt1 self-association. The fraction-L,I,V in the Cdt1 IDR was first calculated over a 5-residue sliding window to determine whether hydrophobic residues were uniformly distributed throughout the sequence or clustered into discrete motifs targetable by mutagenesis (*Figure 4E*, top panel). Overall, the Cdt1 IDR bears an average fraction-L,I,V of 0.19 – or approximately one Leu, Ile, or Val for every five residues – and the hydrophobic residues are generally well distributed throughout the sequence. There are five 5-residue windows where the fraction-L,I,V reaches 0.6, and each third of the IDR (residues 1–100, 101–200, and 201–297) contains at least one of these regions. A fraction-L,I,V = 0.8 occurs only once, between residues 244 and 248. Interestingly, the C-terminal third of the Cdt1 IDR (residues 201–297) contributes more strongly to PEG-induced LLPS than either the first (1–100) or middle third (101–200) of the sequence (*Figure 3E*), suggesting that the 244–248 sequence may contribute to the greater cohesiveness of this region. A higher-precision analysis of Cdt1's hydrophobic residue distribution showed that 67% of Leu, Ile, and Val residues reside in isolation, 22% are immediately adjacent to another Leu, Ile, or Val residue, and 11% are within tripeptides composed of the three amino acids.

Because sequence analysis shows that hydrophobic residues are broadly distributed across the length of the Cdt1 IDR, we were interested to test whether these amino acids contribute to PEG-induced LLPS in an accumulative manner. Two Cdt1 IDR mutants were constructed to probe this issue, one with half of all Ile and Leu residues mutated to alanine (Cdt1$^{Ile/Leu(0.5)}$) and another with all Ile and Leu residues mutated (Cdt1$^{Ile/Leu(1.0)}$). Ile and Leu were chosen as targets, as these are the two most highly enriched hydrophobic residues in initiator IDRs compared to disordered sequences across the proteome. To assess phase separation potential, PEG was titrated (from 0% to 12.5% in 2.5% increments) against each mutant and their disappearance monitored from the supernatant by depletion assay (*Figure 4F*). The phase separation propensity of both variants was significantly disrupted. At the highest concentration of PEG (12.5%), we observed only a modest, insignificant loss of Cdt1$^{Ile/Leu(0.5)}$ and Cdt1$^{Ile/Leu(1.0)}$ (27% and 19%, respectively) from the supernatant, compared to nearly a 100% loss for wild-type Cdt1 (*Figure 3B*). These data establish that, relative to the average disordered domain, initiator IDRs are enriched in branched hydrophobic residues and these amino acid types mediate cohesive inter-IDR interactions that promote phase separation by Cdt1 under conditions of molecular crowding.

## Inter-IDR hydrophobic interactions are necessary for DNA-induced LLPS

Our results have shown that both DNA-bridging interactions (mediated by electrostatics) and inter-IDR interactions (mediated by hydrophobic residues) are important for phase separation by Cdt1. However, the extent to which these two types of interactions synergize remained unclear. We therefore set out to determine whether inter-IDR interactions promote phase separation by Cdt1 in the presence of DNA. During the purification of Cdt1$^{Ile/Leu(0.5)}$ and Cdt1$^{Ile/Leu(1.0)}$ we noticed that both proteins showed a normal affinity for heparin (*Figure 5—figure supplement 1*), indicating that the mutants might bind DNA with wild-type-like affinity. This assumption was confirmed by electrophoretic mobility shift assay, which showed that the DNA-binding affinity of Cdt1$^{Ile/Leu(0.5)}$ ($K_{d, app}$ = 67 nM) and Cdt1$^{Ile/Leu(1.0)}$ ($K_{d, app}$ = 106 nM) is similar to that of the wild-type protein ($K_{d, app}$ = 69 nM) (*Figure 5A*). These data establish Cdt1$^{Ile/Leu(0.5)}$ and Cdt1$^{Ile/Leu(1.0)}$ as separation-of-function mutants that are capable of binding DNA (*Figure 5A*) but that are deficient in their ability to participate in inter-IDR interactions (*Figure 4*).

We next utilized these mutants to ask whether DNA bridging by itself is sufficient to drive Cdt1 LLPS, or whether inter-IDR cohesive interactions are also required. We first used the depletion assay to compare DNA-induced phase separation of Cdt1$^{Ile/Leu(0.5)}$ and Cdt1$^{Ile/Leu(1.0)}$ versus the wild-type protein (*Figure 5B*). While wild-type Cdt1 showed near complete partitioning into the condensed phase (8% protein retention in the supernatant), 63% of Cdt1$^{Ile/Leu(0.5)}$, and 73% of Cdt1$^{Ile/Leu(1.0)}$ remained in the supernatant, demonstrating a marked reduction in DNA-induced phase separation (*Figure 5C*).

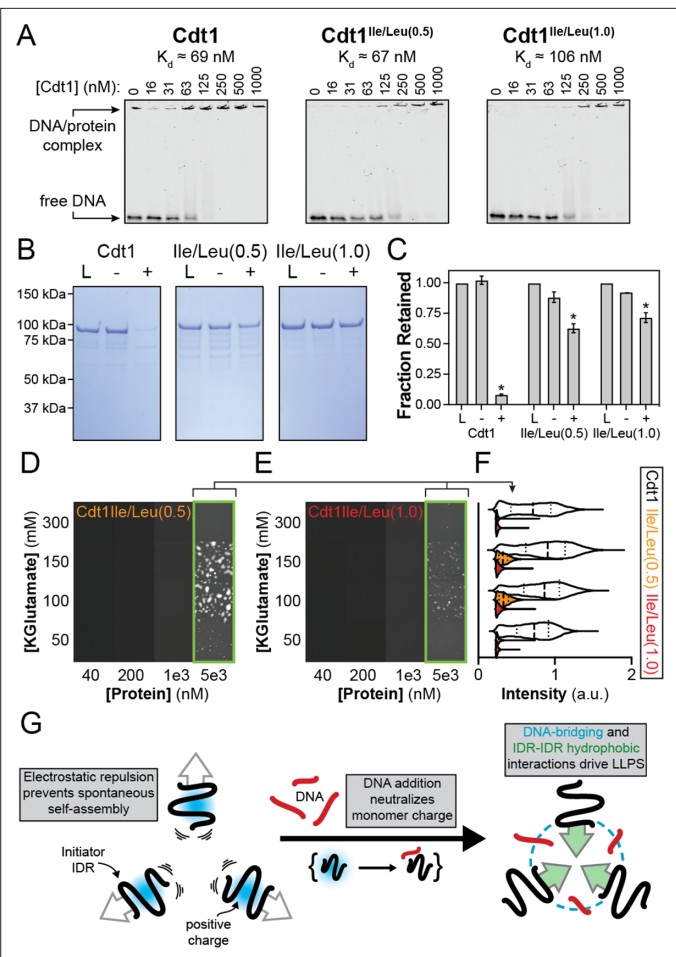

**Figure 5.** Homotypic inter-intrinsically disordered region (IDR) interactions underlie DNA-induced initiator phase separation. (**A**) An electrophoretic mobility shift assay (EMSA) was used to assess the binding affinity of Cdt1 (left), Cdt1$^{Ile/Leu(0.5)}$ (middle) and Cdt1$^{Ile/Leu(1.0)}$ (right) for double-stranded DNA (dsDNA). Binding was assessed with 10 nM IRDye800-labeled dsDNA and a titration of Cdt1 from 16 to 1000 nM. Each variant showed comparable affinity to wild-type Cdt1. (**B**) Depletion assay results assessing DNA-induced phase separation of Cdt1 (left), Cdt1$^{Ile/Leu(0.5)}$ (middle), and Cdt1$^{Ile/Leu(1.0)}$ (right). 'L' is a load control, '-' is in the absence of DNA, and '+' is in the presence of DNA. (**C**) Quantitation of the depletion assay results presented in (**B**). (**D-E**) Phase diagram for stoichiometric amounts of Cdt1$^{Ile/Leu(0.5)}$ (**D**) or Cdt1$^{Ile/Leu(1.0)}$ (**E**) and Cy5-dsDNA at variable protein and KGlutamate concentrations. Tiles bordered in green show conditions where phase separation was observed. Each tile is 75 × 75 μm. (**F**) Violin plot representation of the mean droplet intensity of Cy5-dsDNA at 5 μM protein concentration and variable KGlutamate concentrations for wild-type (white), Cdt1$^{Ile/Leu(0.5)}$ (orange), or Cdt1$^{Ile/Leu(1.0)}$ (red). (**G**) Our data support a model where, in the absence of DNA, inter-IDR electrostatic repulsion prevents spontaneous phase separation. The addition of DNA and resulting IDR-DNA interactions neutralize the initiator IDR charge which drives self-assembly through synergistic heteromeric DNA-bridging interactions and homomeric IDR-IDR hydrophobic interactions. *p < 0.05, *t*-test. Gel and microscopy images are representative from three independent experiments.

The online version of this article includes the following figure supplement(s) for figure 5:

**Figure supplement 1.** Heparin-binding propensity of Cdt1 and the Cdt1 branched hydrophobic residue variants, Cdt1$^{Ile/Leu(0.5)}$ and Cdt1$^{Ile/Leu(1.0)}$.

**Figure supplement 2.** Analysis of Cdt1 intrinsically disordered region (IDR) conservation among *Drosophilidaes* and metazoans.

**Figure supplement 3.** Heatmap analysis of the amino acid composition of initiator intrinsically disordered regions (IDRs; Cdt1, Orc1, and Cdc6) and of an internal intrinsically disordered region (IDR) from Chiffon, the fly homolog of Dbf4.

To gain a quantitative understanding of the DNA-dependent phase separation propensity of these mutants, we generated phase diagrams for Cdt1$^{Ile/Leu(0.5)}$ (*Figure 5D*) and Cdt1$^{Ile/Leu(1.0)}$ (*Figure 5E*) to compare with wild-type protein (*Figure 2F*). We observed a 25-fold reduction in the critical concentration for both mutants. Furthermore, the droplets which formed at 5 µM Cdt1$^{Ile/Leu(0.5)}$ and Cdt1$^{Ile/Leu(1.0)}$ and [KGlutamate] = 150 mM showed a greater than 60% reduction in mean droplet Cy5-dsDNA signal intensity compared to that of the wild-type protein (*Figure 5F*). Overall, these data reveal a striking reduction in the ability of Cdt1$^{Ile/Leu(0.5)}$ and Cdt1$^{Ile/Leu(1.0)}$ to undergo DNA-induced phase separation, despite possessing apparently normal DNA-binding activity, demonstrating that cohesive inter-IDR interactions play a uniquely critical role in driving the phase separation propensity of Cdt1.

## Discussion

In previous work, we discovered that several factors used to initiate DNA replication in metazoans – in particular the Orc1 subunit of ORC, as well as Cdc6 and Cdt1 – possess an N-terminal IDR that promotes protein phase separation upon binding DNA (*Parker et al., 2019*). When these initiator proteins are combined and DNA is added, they form comingled condensates that exclude noncognate phase-separating proteins and are active for the ATP-dependent recruitment of the Mcm2–7 helicase. The biophysical mechanisms underlying LLPS by replication initiators have remained unknown.

Here, we have used *Drosophila* Cdt1 as a model system to dissect the molecular basis for DNA-dependent phase separation by initiation factors. Our studies show that this condensation mechanism has several distinctive features compared to other proteins characterized to date that also undergo LLPS. We find that two types of intermolecular interactions synergize with each other to drive phase separation by Cdt1. One set of interactions occur between Cdt1 and DNA. These associations appear to be nonspecific and electrostatic in nature, with DNA serving as a polyanion that bridges the cationic IDRs of multiple Cdt1 molecules (*Figures 2 and 5G*). A second set of interactions that occur between the IDRs of different Cdt1 protomers were also identified; these appear to be salt-insensitive and primarily hydrophobic in nature (*Figures 4 and 5G*). Importantly, we find that inter-IDR interactions can drive Cdt1, ORC, and Cdc6 LLPS in the absence of DNA (*Figure 3* and *Figure 4—figure supplement 1*). When we abolish these interactions in Cdt1, DNA is insufficient to induce phase separation (*Figure 5*). Thus, inter-IDR interactions provide the primary force behind initiator LLPS, with electrostatic DNA-bridging interactions contributing an additional adhesive force.

In the absence of a crowding reagent, DNA is required to induce initiator phase separation at physiological salt and protein concentrations (*Parker et al., 2019*). This dependency suggests that DNA serves as an essential nucleating factor that determines where Cdt1 can form condensates, ensuring that initiator LLPS is restricted to a chromatin context. Interestingly, other anionic biopolymers, such as RNA and heparin, are also capable of inducing phase separation by Cdt1. This type of scaffold 'promiscuity' has been observed in noninitiator systems that undergo LLPS, such as for the nucleolar protein fibrillarin (*Feric et al., 2016*). Given that the cytosol contains multiple anionic biopolymers, future work will need to answer how replication initiators are specifically targeted to chromatin, a question that is similarly relevant for RNA-dependent cellular bodies (*Berry et al., 2015*; *Elbaum-Garfinkle et al., 2015*; *Feric et al., 2016*; *Guillén-Boixet et al., 2020*; *Mitrea et al., 2016*; *Molliex et al., 2015*; *Van Treeck et al., 2018*). The selection of an appropriate nucleic acid scaffold across different phase-separating systems is likely to rely on both general mechanisms, such as the sequestration of alternative, inappropriate scaffolds by proteins with different or nonexistent LLPS properties, and by system-specific mechanisms, such as a requirement for particular sequences or secondary structure (*Langdon et al., 2018*; *Maharana et al., 2018*).

A growing body of work has demonstrated the importance of aromatic residues in driving protein phase separation (*Chiu et al., 2020*; *Chong et al., 2018*; *Lin et al., 2017*; *Nott et al., 2015*; *Pak et al., 2016*; *Qamar et al., 2018*; *Wang et al., 2018*), along with a related role for π–π and π–cation interactions (*Vernon et al., 2018*). In parallel, many studies have shown that condensates formed through aromatic residue interactions are readily dissolved by 1,6-HD (*Table 1*). Cdt1 LLPS is both resistant to treatment with 1,6-HD and does not rely on aromatic residues within its IDR for phase separation (*Figure 1*). ORC and Cdc6 phase separation is also resistant to 1,6-HD (*Figure 1—figure supplement 2*). Although we have not explicitly investigated the role of aromatic residues within these proteins, we note that the fraction aromatic residues of Orc1 (0.019) and Cdc6 (0.012) is lower than that of Cdt1 (0.027), suggesting that aromaticity is dispensable for LLPS by replication initiation factors in general.

Similarly, charged residues have an established role in facilitating homomeric inter-IDR interactions that lead to phase separation (*Elbaum-Garfinkle et al., 2015*; *Mitrea et al., 2018*; *Nott et al., 2015*), yet the inter-IDR interactions that drive initiator phase separation (*Figure 4* and *Figure 4—figure supplement 1*) are salt insensitive.

Instead of relying on aromatic-mediated π–π/π–cation interactions or charge–charge interactions, our data show that interactions between Cdt1 IDRs are instead facilitated by branched hydrophobic residues. Although hydrophobic sequences have been shown to be important for the phase separation of other factors, such as the extracellular matrix protein elastin (*Quiroz and Chilkoti, 2015*; *Reichheld et al., 2017*; *Yeo et al., 2011*) and the NICD (*Pak et al., 2016*), important mechanistic distinctions exist with respect to metazoan replication initiators. For example, both NICD and Cdt1 bear an identical FCRs (FCR = 31%) and rely on a counterion for phase separation. However, inter-IDR interactions within NICD appear to rely primarily on sequence aromatics (11% aromatic content), with hydrophobic residues playing only a supporting role in the self-association of this protein (*Pak et al., 2016*). By contrast, elastin is more like metazoan replication initiators in that it is relatively devoid of aromatic residues (3.9%) and contains a similar content of branched hydrophobic amino acids that underly phase separation (elastin and Cdt1 fraction-L,I,V = 0.21 and 0.19, respectively); however, elastin also differs from initiators in that it readily forms condensates in the absence of a counterion. Elastin's hydrophobic sequences are additionally contained within highly repetitive 'VPGVG' and 'GLG' sequences, and scrambling or shuffling these motifs negatively impacts elastin phase separation (*Pepe et al., 2008*; *Pepe et al., 2005*; *Toonkool et al., 2001*). For their part, initiator IDRs lack such repeats and (for Cdt1 at least) LLPS is insensitive to sequence order. We speculate that the inter-IDR interactions which occur in initiators are weaker than those of Elastin due to the nonrepetitive, distributed nature of their hydrophobic residues, a property that necessitates additional counterion-mediated interactions to promote Cdt1 self-assembly.

Understanding the distinctive molecular interactions of different phase-separating systems is essential for developing *in silico* approaches for predicting the cellular partitioning of a given IDR from primary sequence alone. Such efforts will ultimately require an understanding of both the specific and general features of condensates' scaffolding components (e.g., repetitive motifs vs. amino acid composition). The present work highlights the importance of the general sequence features of initiator IDRs (as opposed to strict amino acid order) in promoting LLPS. Strikingly, we find that the Cdt1 IDR can be fully scrambled without losing the ability to form condensates (*Figure 3I, J*) and that condensates formed from the scrambled variant can recruit wild-type Cdt1 into droplets (Figure 3—figure supplement 3). Regions of NICD can similarly be scrambled without losing the ability to form cellular condensates (*Pak et al., 2016*), as can the LCDs of certain transcription factors without affecting promoter targeting (*Brodsky et al., 2020*). These studies demonstrate the importance of sequence composition in not only driving phase separation but also in facilitating interactions with partner proteins, and is consistent with recent work showing that proteins with similar functional annotation have disordered sequences with similar, evolutionary conserved physical–chemical features (pI, composition, FCR, kappa, etc.) (*Zarin et al., 2019*). Our observations involving the cocondensation properties of Cdt1, ORC, and Cdc6 – three proteins with IDRs of similar amino acid composition but no direct sequence homology (*Parker et al., 2019*) – provide experimental support for this 'like-recruits-like' concept. Future work will need to address whether heteromeric inter-IDR interactions (e.g., Orc1$^{IDR}$–Cdc$^{IDR}$) are indeed governed strictly by sequence composition or whether linear sequence motifs facilitate specific interactions, as was recently suggested for human Orc1 and Cdc6 (*Hossain et al., 2021*). Further, while metazoan Cdt1 orthologs have no identifiable sequence identity across their disordered domains (*Figure 5—figure supplement 2A, B*), they do have a similar amino acid composition (*Parker et al., 2019*). This led us to predict that the capacity to phase-separate was likely conserved across metazoan initiators and we confirmed this for human Cdt1 (*Parker et al., 2019*). These predictions were further validated for human Orc1 and Cdc6 with the demonstration that these proteins form condensates in the presence of DNA (*Hossain et al., 2021*). These results suggest that certain classes of disordered domains can have conserved functionality in the absence of linear sequence identity.

Although amino acid composition is the primary determinant underpinning LLPS by replication initiators, short sequence motifs can nonetheless play a critical role in modulating condensation behavior and controlling biological function. Sequence information within the Orc1 and Cdt1 IDR's is high, and while saltatory leaps seem to have occurred between phyla (*Figure 5—figure supplement 2B*), the

conservation of sequence homology within the *Drosophila* genus (*Figure 5—figure supplement 2C* and *Parker et al., 2019*), over millions of years, is remarkable and indicates an evolutionary pressure on sequence order for unknown function. A conserved sequence feature within the IDRs of Orc1, Cdt1, and Cdc6 are CDK phosphorylation sites which are known to regulate replication initiation in vivo (*Findeisen et al., 1999*; *Lee et al., 2012*). We have previously shown that the phosphorylation of these sites – which are distributed throughout the initiator IDRs – by CDKs abrogates the ability of the *Drosophila* initiators to form condensates (*Parker et al., 2019*), and a recent finding shows that phosphorylation impedes inter-IDR interactions between human ORC1 and CDC6 as well (*Hossain et al., 2021*). Notably, consensus motifs for CDK-dependent phosphorylation (full site = [S/T]PX[R/K] and minimal site = [S/T]P) are conserved in a majority of metazoan Cdt1s (*Figure 5—figure supplement 2D*) and in all sequenced *Drosophilidae* Cdt1 orthologs (*Figure 5—figure supplement 2E*), leading us to predict that phospho-tunable phase separation is a broadly conserved mechanism for regulating metazoan replication licensing. Given that a scrambled IDR can still support LLPS in vitro through both self-interactions and cross-interactions with a wild-type IDR sequence (*Figure 3*), it will be useful to determine whether such a construct (with or without the native CDK sequences) can support normal kinetics of chromatin association and MCM recruitment and/or cell viability. Such studies, along with deletion and more targeted mutagenesis efforts, will be important to define how the plasticity of IDR sequences can be fine-tuned to elicit specific timing and partnering responses.

Defining the molecular 'grammar' that encodes LLPS in metazoan initiators likewise has implications for understanding the impact of initiator phase separation in vivo. Admittedly, neither this nor our previous study (*Parker et al., 2019*) provides unequivocal evidence for initiator phase separation in cells. It is also unclear what function such concentrated initiator assemblies might have, such as improving the kinetics or efficiency of helicase loading (as we have suggested here and previously [*Parker et al., 2019*]), or something else entirely, such as maintaining the appropriate nuclear levels of the licensing factors (as proposed in *Hossain et al., 2021*). We note that our system lacks the convenience of identifying initiator condensation morphologically (i.e., by the presence of round cellular foci), as mitotic chromosomes – the stage of initiator assembly – are relatively rigid structures built from a central proteinaceous scaffold through an energy-consuming reaction (*Earnshaw and Laemmli, 1983*; *Gibcus et al., 2018*; *Hirano and Mitchison, 1994*; *Kim et al., 2020*). Thus, we predict that initiators condense on the surface of mitotic chromosomes without altering their shape. Consistently, work in multiple model organisms has revealed a switch-like transition in initiator localization during anaphase, at which point initiators begin to coat chromosomes (*Baldinger and Gossen, 2009*; *Kara et al., 2015*; *Parker et al., 2019*; *Sonneville et al., 2012*). We predict that phosphorylation-dependent control of initiator self-assembly underlies this behavior and produces a dense, replication-competent zone of chromatin-bound initiators poised to drive the precipitous, genome-wide loading of the Mcm2–7 complex in late mitosis (*Dimitrova et al., 2002*; *Méndez and Stillman, 2000*; *Okuno et al., 2001*). This moment in the cell cycle is an opportune time to prepare for replication, as conflicts with the transcriptional machinery are minimized (*Palozola et al., 2017*) and the chromatin substrate is, relative to interphase, uniformly compacted (*Ou et al., 2017*). The identification of Cdt1 mutants that selectively block phase separation, but not DNA binding (*Figure 5*), affords an opportunity to directly investigate the physical nature of these global initiator interactions in vivo.

Beyond helicase loading and licensing, initiator LLPS might also serve to prime replisome assembly. Following helicase loading, Mcm2–7 is phosphorylated by the Dbf4-dependent kinase, a heterodimer of Cdc7 and Dbf4 (*Deegan et al., 2016*; *Sheu and Stillman, 2010*; *Yeeles et al., 2015*). Intriguingly, the *Drosophila* homolog of Dbf4, Chiffon, contains multiple IDRs, including a large internal IDR (residues 903–1209) that possesses striking compositional similarity to initiator-type IDRs (*Figure 5—figure supplement 3*). The identification of an IDR in Chiffon with compositional similarity to the initiator-type IDRs suggests that such sequences may be present in chromatin-associated factors beyond replication licensing components. Indeed, the capacity to undergo DNA-dependent phase separation may have broad utility for chromatin-localized factors and their respective cellular pathways. Thus, our future work will aim to develop algorithms that accurately identify proteins proteome wide that possess disordered domains with compositional homology to initiator IDRs and to understand how these sequences impact protein functional dynamics.

In summary, the present work provides a detailed view of the DNA-dependent LLPS mechanism for *Drosophila* Cdt1. Due to a high level of IDR compositional homology and an ability to form comingled

phases (*Parker et al., 2019*), the mechanism we describe for Cdt1 phase separation likely extends to the other replication initiation factors, ORC and Cdc6. These studies set the stage for investigating the physiological significance of initiator self-assembly in replication licensing and for identifying other sequences in the proteome that possess IDRs capable of associating with chromatin and initiation factors alike.

# Materials and methods

## Key resources table

| Reagent type (species) or resource | Designation | Source or reference | Identifiers | Additional information |
|---|---|---|---|---|
| Recombinant DNA reagent | 2Cc-T | QB3 Macrolab (UC Berkeley) | RRID:Addgene_37237 | Ligation independent cloning (LIC); *E. coli* expression vector |
| Recombinant DNA reagent | 1GFP | QB3 Macrolab (UC Berkeley) | RRID:Addgene_29663 | Ligation independent cloning (LIC); *E. coli* expression vector |
| Peptide, recombinant protein | TEV | QB3 Macrolab (UC Berkeley) | | Used at 1/10 (wt/wt) TEV/substrate ratio |
| Strain, strain background (*Escherichia coli*) | Rosetta 2(DE3) pLysS | QB3 Macrolab (UC Berkeley) | | Chemically competent cells |

## Protein production and purification

Wild-type and mutant Cdt1 coding sequences (*Table 3*) were cloned into the QB3 Macrolab vector 2Cc-T to append a tobacco etch virus (TEV) protease-cleavable C-terminal hexa-histidine (His6)-maltose-binding protein (MBP) tag. To produce eGFP-tagged Cdt1, the wild-type Cdt1 coding sequence was cloned into QB3 Macrolab vector 1GFP to append an N-terminal His6-eGFP tag. All Cdt1 variants were produced from Rosetta 2(DE3) pLysS cells. Cells were grown in 2xYT broth at 37°C to an $OD_{600}$ = 0.8 and after a 15-min incubation in an ice bath induced with 1 mM isopropyl β-D-thiogalactoside (IPTG). After growth overnight at 16°C, cells were harvested by centrifugation and cell pellets frozen at −80°C until further processing.

Cell pellets were resuspended in Lysis Buffer (20 mM Tris, pH 7.5, 500 mM NaCl, 30 mM imidazole, 10% glycerol, 200 µM PMSF, 1× cOmplete EDTA-free Protease Inhibitor Cocktail (Sigma-Aldrich), 1 mM 2-mercaptoethanol (BME), and 0.1 mg/ml lysozyme) and lysed by sonication. Lysates were clarified by centrifugation at 30,000 × *g* for 1 hr and filtered through a 0.45 µm bottle-top filter unit (Nalgene Rapid-Flow, Thermo Fisher). Lysates were then passed over a 5 ml HisTrap HP column (GE Healthcare) and washed with 10 column volumes (CVs) of Nickel Wash Buffer (20 mM Tris, pH 7.5, 1 M NaCl, 30 mM imidazole, 10% glycerol, 200 µM PMSF, and 1 mM BME) followed by five CVs of Low Salt Nickel Wash Buffer (20 mM Tris, pH 7.5, 150 mM NaCl, 30 mM imidazole, 10% glycerol, 200 µM PMSF, and 1 mM BME). Protein was eluted from the column with five CVs of Nickel Elution Buffer (20 mM Tris, pH 7.5, 150 mM NaCl, 500 mM imidazole, 10% glycerol, 200 µM PMSF, and 1 mM BME). Eluted protein was then loaded onto a 5-ml HiTrap Heparin HP column (GE Healthcare), washed with five CVs of Heparin Wash Buffer (20 mM Tris, pH 7.5, 150 mM NaCl, 10% glycerol, 200 µM PMSF, and 1 mM BME) and eluted with a linear gradient of increasing salt from 150 mM to 1 M NaCl (20 mM Tris, pH 7.5, 10% glycerol, 200 µM PMSF, and 1 mM BME). Fractions containing Cdt1 were pooled and TEV protease (QB3 Macrolab) added at a 1:20 (wt/wt) ratio of protein:TEV and incubated overnight at 4°C. A second nickel affinity step was used to remove TEV, uncleaved protein, and the free His6-MBP tag. Finally, the sample was concentrated to 2 ml and loaded onto a HiPrep 16/60 Sephacryl S-300 HR column (GE Healthcare) equilibrated and run in Sizing Buffer (50 mM HEPES [4-(2-hydroxyethyl)-1-piperazineethanesulfonic acid], pH 7.5, 300 mM KGlutamate, 10% glycerol, and 1 mM BME). Peak fractions were collected, concentrated in a 30 K Amicon Ultra-15 concentrator (Millipore), flash frozen in liquid nitrogen, and stored at −80°C. The same protocol was used to purify GFP-tagged Cdt1 with the exception that TEV protease was not added, and the second nickel affinity step was omitted.

*D. melanogaster* ORC and Cdc6 were purified as previously described (*Parker et al., 2019*).

**Table 3.** IDR sequences of the proteins used in this study.
All IDR sequences were fused back to wild-type Cdt1 residues 298–743.

| | |
|---|---|
| *Dm*Cdt1 (WT) | ATGGCCCAGCCATCGGTAGCTGCCATTTTTCACAAACCGCAAACGCGCCGCCTTGGATGATGCTATCAGTATCAAGAACAGGCGTTTGGTGGAACCGCTGAAACCGTCTCTCCTGCCTCCGCCCCTTCCCAGTTGCCAGCCGGCGACCAGGATGCGGATCTAGACACCCTGAAGGCGGCGGCCGCACGGGCATGCGTACCCGATCCGGACGCACTGCCCGACTAATTGTCACCGCCGCTCAAGAGAGCAAAAAGAAGACACCGGCTGCCGCCAAGATGGAGCCACACATCAAGCAGCCCAAGCTGGTGCAATTCATTAAAAAGGGCACTCTGTCGCCCAGGAAACAGGCTCAGTCCAGTAAGCTGGACGAAGGAGGAGCTGCAGCAGTCGTGCAGCTATCGCCGAGAAACCGGACGGGATGCCAAGATGAGCGTGCATCCGAGAACGGCAAACATGAGCCCAAGGTTAACTTCACCATCACAAGCCAGCAGAATGCGGACAATGTGCAGCGTGGCCTGCGCACACCCACCAAGCAGATCCTCAAGGATGCCTCGCCGATCAAGGCCGGATCTCCGCCGTCAGCTCACTTTGACGAGGTAAAAACGAAGGTATCGCGGAGTGCCAAGCTGCAGGAACTCAAGGCAGTGCTGGCCCTTAAGGCGGCGCTCGAGCAGAAGCGCAAGGAGCAGGAGGAGCGCAACAGGAAACTCCGCGACGCTGGCCCCTCCCCATCGAAGTCCAAGATGAGCGTCAAGGAATTCGACACAATCGAACTGGAGGTGCTTATAAGCCCTTTGAAGACCTTCAAGACTCCCACAAAAATACCGCCCACCCACCCCGGACAAACATGAGCTTATGTCGCCGCGTCACACTGACGTCTCCAAGCGCCTTCTCTTCAGTCCGGCCAAAAATGGATCTCCTGTCAAATTGGTGGAG<br><br>MAQPSVAAFFTNRKRAALDDAISIKNRRLVEPAETVSPASAPSQLPAGDQDADLDTLKAAATGMRTRSGRTARLIVTAAQESKKKTPAAAKMEPHIKQPKLVQFIKKGTLSPRKQAQSSKLDEEELQQSSAISEHTPKVNFTITSQQNADNVRQRGLRTPTKQILKDASPIKADLRRQLTFDEVKTKVSRSAKLQELKAVLALKAALEQKRKEQEERNRKLRDAGPSPSKSKMSVQLKEFDTIELEVLISPLKFTKTPTKIPPPTPDKHELMSPRHTDVSKRLLFSPAKNGSPVKLVE |
| *Dm*Cdt1 Phe→Leu | ATGGCACAGCCTAGCGTCGCCGCACTTTTAACCAATCGTAAGCGTGCCGCCTTAGATGACGCCATTTCGATTAAGAATCGCCGTCTGGTTGAGCCCGCGGAAACAGTCAGCCCGGCAAGTGCCCCGTCGCAGTTGCCCGCCGGCGATCAGGACGCAGATTTAGATACATTAAAAGCTGCGGCCACAGGTATGCGTACCCGTTCTGGGCGTACGGCCCGTTTGATTGTGACAGCGGCCCAAGAGTCGAAAAAAAAGACCCCAGCTGCTGCTAAAATGGAACCACATATCAAGCAACCGAAACTGGTCCAGTTAATCAAAAAAGGAACCCTGAGCCCGCGTAAACAAGCTCAGTCAAGTAAGTTGGATGAGGAAGAGTTGCAGCAGTCGTCAGCCATCTCCGAACACACCCCCAAGGTGAATCTGACCATCACTAGTCAGCAAAACGCGGATAACGTTCAGCGCGGACTGCGTACCCCCACCAAACAAATTTTGAAGGATGCCTCTCCAATTAAGGCAGATCTGCGTCGTCAACGTGACGTTAGACAGAGGTAAAGACCAAAGTCAGCTCGCTCAGCCAAGCTGCAGGAACTGAAGGCTGTATTAGCTCTGAAGGCAGCCTTGGAACAGAAGCGCAAGGAGCAAGAAGAACGCAACCGCAAGCTGCGCGACGCCGGCCCGTCGCCAAGCAAGTCTAAGATGTCGGTACAACTTAAAGAATTAGACACTATTGAGTTAGAGGTCCTTATCTCTCCGCTGAAAACGTTGAAGACCCCCACTAAGATTCCGCCGCCAACCCCTGATAAACACGAGTTAATGTCTCCCCGTCATACCGACGTATCGAAAGCGCCTTTTATTGAGTCCTGCGAAGAACGGATCCCCAGTAAAATTGGTCGAG<br><br>MAQPSVAALLTNRKRAALDDAISIKNRRLVEPAETVSPASAPSQLPAGDQDADLDTLKAAATGMRTRSGRTARLIVTAAQESKKKTPAAAKMEPHIKQPKLVELTITSQQQNADNVRQRGLRTPTKQILKDASPIKADLRRQLTLDEVKTKVSRSAKLQELKAVLALKAALEQKRKEQEERNRKLRDAGPSPSKSKMSVQLKELDTIELEVLISPLKTLKTPTKIPPPTPDKHELMSPRHTDVSKRLLLSPAKNGSPVKLVE |
| *Dm*Cdt1 Phe→Ala | ATGGCACAGCCCTCGGTAGCGGCAGCCGCCACGAATCGCAAGCGTGCCGCTTAGACGATGCAATCAGTATTAAGAATCGCCGTCTTGTAGAGCCTGCGGAGACGGTCTCTCCTGCAAGTGCTCCTTCTCAGCTTCCAGCGGGAGATCAGGACGCGACTTAGATACACTGAAGGCTGCCGCAACCGGAATGCGTACACGTTCAGGTCGCACTGCCCGTCTTATCGTGACAGCCGCCCAGGAGTCTAAAAAAAAAACTCCAGCAGCAGCAAAGATGGAACCTCATATCAAGCAGCCAAAACTGGTACAGGCGATCAAAAAGGGCACTCTGTCCCCACGCAAGCAAGCTCAATCGAGCAAGTTAGACGAGGAGGAGTTGCAGCAGTCTAGCGCCATTTCCGAACATACTCCAAAGGTAAACGCGACGATCACTTCTCAACAAAATGCCGATAATGTCCAGCGTGGGTTACGCACGCCTACAAAACAGATTTTGAAGGACGCTAGCCCTATTAAAGCCGATCTTCGTCGTCAACTGACCGGCTGATGAAGTGAAGACCAAGGTAAGCCGTTCCGCAAAACTGCAAGAACTTAAAGCCGTGCTGGCTTTAAAAGCGGCTTTGGAACAAAAGCGTAAAGAGCAGGAAGAGCGTAACCGTAAGTTGCGTGACGCAGGACCAAGCCCTTCAAAGTCCAAGATGAGCGTACAACTTAAGGAAGCTGATACTATTGAATTGGAGGTACTGATCTCGCCTCTTAAGACTGCCAAGACGCCCACAAAAATCCCGCCCCCCACACCTGACAAGCATGAACTTATGAGCCCACGCCACACGGATGTGTCCAAGCGTTTGCTGGCCAGCCCCGCTAAGAACGGATCCCCGGTCAAACTGGTAGAG<br><br>MAQPSVAAAATNRKRAALDDAISIKNRRLVEPAETVSPASAPSQLPAGDQDADLDTLKAAATGMRTRSGRTARLIVTAAQESKKKTPAAAKMEPHIKQPKLVQAIKKGTLSPRKQAQSSKLDEEELQQSSAISEHTPKVNATITSQQNADNVRQRGLRTPTKQILKDASPIKADLRRQLTADEVKTKVSRSAKLQELKAVLALKAALEQKRKEQEERNRKLRDAGPSPSKSKMSVQLKEADTIELEVLISPLKTAKTPTKIPPPTPDKHELMSPRHTDVSKRLLASPAKNGSPVKLVE |
| *Dm*Cdt1 Uniform | ATGGCTCAACCTTCTGTCAAAGCTGCATTCTTTACCCGCGAAAACAAGGCGGCCGGACTTGGCGATCTCAAAGATTAACTTGCGCGAAGTGCCCGCGAAAACTGTTTCTCCAGACGCTTCAAAAGCTCGTGAACCTAGTCAACTTCCGAAAGCAGGACAGGCTTTGACTGACCGTGAGAAACTGGCGGCAGCCACAGGCAAGATGACTTCGGGTCGCGAAACGGCCGATAAGTTAATCGTAACGGCTGCCAAACAATCACGTGAAACACCCGCCGCAAAGGCAGATATGCCGCATATTCAGAAGCGCGAACCCTTGGTGCAGTTCATTAAGGGGACCTTAGACTCCCCTCGTGAGCAAAAAGCCCAAGTCTTCTCTTTTGAAACAACAGTCGCGTGAAGATTCAGCCATTAAGTCACATACGCCCGTGAACAAATTCCGCGAAACAATCACAAGTGACCAAAAGCAAAACGCCAACGTACAGCGTGAGAAGGGCTTGACGCCGACACAAAAAGGACATTTTAGCCTCACGCGAACCTATCAAGGCCTTGCAACTGACTTTTAAGGTGACTGATCGTGAAGTCTCATCTGCGAAACTTCAATTAGCGGTTCTGAAGCGCGAAGCCTTAGATGCAGCTCTGCAAAAACAAAACCTTGCAAGGTCGCCAAAGTCCCAGTAGTGACATGTCGAAGTTCAGTTACGTGAAGTTCGACAATCAAACTGGTTTTGATTTCACCAGACAAGTTGCGTGAAACTTTCACTCCAACCAAAATTCCGCCACCCACGCCCCGCGAGGATAAGCATTTAATGAGTCCCCACAAAACCGTTTCACTGCGCGAACTGTTCAAATCCGACCCTGCTAATGGAAGTAAGCCGGTACGCGAGCTTGTC<br><br>MAQPSVKAAFFTRENKAADLAISKINLREVPAKTVSPDASKAREPSQLPKAGQALTDREKLAAATGKMTSGRETADKLIVTAAKQSRETPAAKADMPHIQKREPLVQFIKGTLDSPREQKAQSSLLKQQSREDSAIKSHTPVNKFRETITSDQKQNANVQREKGLTPTQKDILASREPIKALQLTFKVTDREVSSAKLQLAVLKREALDAALQKQNLAGREPKSPSSDMSKVQLREFTIKLVLISPDKLRETFTPTKIPPPTPREDKHLMSPHKTVSLRELFKSDPANGSKPVRELV |
| *Dm*Cdt1 Δ1–100 | ATGCTGGTGCAATTCATTAAAAAGGGCACTCTGTCGCCCAGGAAACAGGCTCAGTCCAGTAAGCTGGACGAAGGAGGAGCTGCAGCAGTCGTCGGCCATAAGCGAGCACACGCCCAAGGTTAACTTCACCATCACAAGCCAGCAGAATGCGGACAATGTGCAGCGTGGCCTGCGCACACCCACCAAGCAGATCCTCAAGGATGCCTCGCCGATCAAGGCCGGATCTCCGCCGTCAGCTCACTTTGACGAGGTAAAAACGAAGGTATCGCGGAGTGCCAAGCTGCAGGAACTCAAGGCAGTGCTGGCCCTTAAGGCGGCGCTCGAGCAGAAGCGCAAGGAGCAGGAGGAGCGCAACAGGAAACTCCGCGACGCTGGCCCCTCCCCATCGAAGTCCAAGATGAGCGTGCAGCTCAAGGAATTCGACACAATCGAACTGGAGGTGCTTATAAGCCCTTTGAAGACCTTCAAGACTCCCACAAAAATACCGCCCACCCACCCCGGACAAACATGAGCTTATGTCGCCGCGTCACACTGACGTCTCCAAGCGCCTTCTCTTCAGTCCGGCCAAAAATGGATCTCCTGTCAAATTGGTGGAG<br><br>MLVQFIKKGTLSPRKQAQSSKLDEEELQQSSAISEHTPKVNFTITSQQNADNVRQRGLRTPTKQILKDASPIKADLRRQLTFDEVKTKVSRSAKLQELKAVLALKAALEQKRKEQEERNRKLRDAGPSPSKSKMSVQLKEFDTIELEVLISPLKFTKTPTKIPPPTPDKHELMSPRHTDVSKRLLFSPAKNGSPVKLVE |
| *Dm*Cdt1 Δ101–200 | ATGGCCCAGCCATCGGTAGCTGCCATTTTTCACAAACCGCAAACGCGCCGCCTTGGATGATGCTATCAGTATCAAGAACAGGCGTTTGGTGGAACCGCTGAAACCGTCTCTCCTGCCTCCGCCCCTTCCCAGTTGCCAGCCGGCGACCAGGATGCGGATCTAGACACCCTGAAGGCGGCGGCCGCACGGGCATGCGTACCCGATCCGGACGCACTGCCCGACTAATTGTCACCGCCGCTCAAGAGAGCAAAAAGAAGACACCGGCTGCCGCCAAGATGGAGCCACACATCAAGCAGCCCAAGGCCCTTAAGGCGGCGCTCGAGCAGAAGCGCAAGGAGCAGGAGGAGCGCAACAGGAAACTCCGCGACGCTGGCCCCTCCCCATCGAAGTCCAAGATGAGCGTGCAGCTCAAGGAATTCGACACAATCGAACTGGAGGTGCTTATAAGCCCTTTGAAGACCTTCAAGACTCCCACAAAAATACCGCCCACCCACCCCGGACAAACATGAGCTTATGTCGCCGCGTCACACTGACGTCTCCAAGCGCCTTCTCTTCAGTCCGGCCAAAAATGGATCTCCTGTCAAATTGGTGGAG<br><br>MAQPSVAAFFTNRKRAALDDAISIKNRRLVEPAETVSPASAPSQLPAGDQDADLDTLKAAATGMRTRSGRTARLIVTAAQESKKKTPAAAKMEPHIKQPKALKAALEQKRKEQEERNRKLRDAGPSPSKSKMSVQLKEFDTIELEVLISPLKFTKTPTKIPPPTPDKHELMSPRHTDVSKRLLFSPAKNGSPVKLVE |
| *Dm*Cdt1 Δ201–297 | ATGGCCCAGCCATCGGTAGCTGCCATTTTTCACAAACCGCAAACGCGCCGCCTTGGATGATGCTATCAGTATCAAGAACAGGCGTTTGGTGGAACCGCTGAAACCGTCTCTCCTGCCTCCGCCCCTTCCCAGTTGCCAGCCGGCGACCAGGATGCGGATCTAGACACCCTGAAGGCGGCGGCCGCACGGGCATGCGTACCCGATCCGGACGCACTGCCCGACTAATTGTCACCGCCGCTCAAGAGAGCAAAAAGAAGACACCGGCTGCCGCCAAGATGGAGCCACACATCAAGCAGCCCAAGCTGGTGCAATTCATTAAAAAGGGCACTCTGTCGCCCAGGAAACAGGCTCAGTCCAGTAAGCTGGACGAAGGAGGAGCTGCAGCAGTCGTCGGCCATAAGCGAGCACACGCCCAAGGTTAACTTCACCATCACAAGCCAGCAGAATGCGGACAATGTGCAGCGTGGCCTGCGCACACCCACCAAGCAGATCCTCAAGGATGCCTCGCCGATCAAGGCCGGATCTCCGCCGTCAGCTCACTTTGACGAGGTAAAAACGAAGGTATCGCGGAGTGCCAAGCTGCAGGAACTCAAGGCAGTGCTG<br><br>MAQPSVAAFFTNRKRAALDDAISIKNRRLVEPAETVSPASAPSQLPAGDQDADLDTLKAAATGMRTRSGRTARLIVTAAQESKKKTPAAAKMEPHIKQPKLVQFIKKGTLSPRKQAQSSKLDEEELQQSSAISEHTPKVNFTITSQQNADNVRQRGLRTPTKQILKDASPIKADLRRQLTFDEVKTKVSRSAKLQELKAVL |
| *Dm*Cdt1 ΔCoiled-coil | ATGGCCCAGCCATCGGTAGCTGCCATTTTTCACAAACCGCAAACGCGCCGCCTTGGATGATGCTATCAGTATCAAGAACAGGCGTTTGGTGGAACCGCTGAAACCGTCTCTCCTGCCTCCGCCCCTTCCCAGTTGCCAGCCGGCGACCAGGATGCGGATCTAGACACCCTGAAGGCGGCGGCCGCACGGGCATGCGTACCCGATCCGGACGCACTGCCCGACTAATTGTCACCGCCGCTCAAGAGAGCAAAAAGAAGACACCGGCTGCCGCCAAGATGGAGCCACACATCAAGCAGCCCAAGCTGGTGCAATTCATTAAAAAGGGCACTCTGTCGCCCAGGAAACAGGCTCAGTCCAGTAAGCTGGACGAAGGAGGAGCTGCAGCAGTCGTCGGCCATAAGCGAGCACACGCCCAAGGTTAACTTCACCATCACAAGCCAGCAGAATGCGGACAATGTGCAGCGTGGCCTGCGCACACCCACCAAGCAGATCCTCAAGGATGCCTCGCCGATCAAGGCCGGATCTCCGCCGTCAGCTCACTTTGACGAGGTAAAAACGAAGGTATCGCGGAGTGCCAAGCTGCAGGAAGGCCCCTCCCCATCGAAGTCCAAGATGAGCGTGCAGCTCAAGGAATTCGACACAATCGAACTGGAGGTGCTTATAAGCCCTTTGAAGACCTTCAAGACTCCCACAAAAATACCGCCCACCCACCCCGGACAAACATGAGCTTATGTCGCCGCGTCACACTGACGTCTCCAAGCGCCTTCTCTTCAGTCCGGCCAAAAATGGATCTCCTGTCAAATTGGTGGAG<br><br>MAQPSVAAFFTNRKRAALDDAISIKNRRLVEPAETVSPASAPSQLPAGDQDADLDTLKAAATGMRTRSGRTARLIVTAAQESKKKTPAAAKMEPHIKQPKLVQFIKKGTLSPRKQAQSSKLDEEELQQSSAISEHTPKVNFTITSQQNADNVRQRGLRTPTKQILKDASPIKADLRRQLTFDEVKTKVSRSAKLQEGPSPSKSKMSVQLKEFDTIELEVLISPLKTFKTPTKIPPPTPDKHELMSPRHTDVSKRLLFSPAKNGSPVKLVE |
| *Dm*Cdt1 ScrIDR | ATGTTAACAGAGGATTTGCAAAAGTTTACCCCGCCAAAGCGCGGGATTACGACAAGTCTTGAAAAAACGAGTCCCGCCGCCTCACAGAAACCTATCTCGCAATTGCCCAAGGATACTAAAAAGAAGGCACCGCCGCACGGGAGCAAGAGGCCGAAGTCAAAGCACCAGGCCGCGAACTTGAGAAAGATCCTCAGCTTGACTCAATCCGTCGCTTGGTAAACGCCCCGAAGGTCAAAATTTCACAACAGGTCGTGGCGGGGAGAGTTAGAGGCTGGGAAGGAAGCAGGTTTGACCGCTGTTAAGGACAAAAGCGGCACAGGAAAGCCCGCCGTTCACCCAAGGATTCGCTGAATTTGACATCCGGCTCGGCGCTCAGCCGTCGCGAGCGTGAGTAAGTTATCCCCGCTGATTGATGCAAAGCTGCTGGAAACTTTATTAGATACAAAGACCGCCCGTACTGCCCCGGCCATTAAGCCGCGTATGAACAAGCAGCCACGCGACCATAAGACCCAACTGACGCAGCGCCGCCAAGGAGCCGGCCATCCGCAACATGAGCCTGCTAGATGATAAGTTTGCCTTGGCCCTGAGCAAGGTCAAGATCCATCTGAGTTTCGTAATTCCCTGCCCGTAGTCAAGTCCTCCGTAGTCTTAGACATCGCG<br><br>MLTEDLQKFTPPKRGITTSLEKTSPAASQKPISQLPKDTKKKAPPTEQEAEVKAPGRELEKDPQLDSIRRLVNAPKVKISQQRRQAVAAGELEAGKEAGLTAVKDKGAEEARRSPKDSLNFDIRSRSAVASVSKLSPLIDAKLLETLLDTKTARTAPAIKPRMNKQPRDHKTQQTRRQPKEPAIRTRMKAIQAEAATHSSRKQFTFKGEVELPKSKQKLATANPAQVKTQMLSDLSEPEHTMLTKNSPARTIQVSQFSRPNFLVDDKFALALSKVKIHLSFDNSLPVVKSSLVLDIA |
| *Dm*Cdt1 Ile/Leu(0.5) | ATGGCCCAGCCCTCCGTTGCTGCGTTTTTTACAAACCGCAAGCGTGCTGCAGCGGATGATGCCGCGAGTATCAAGAACCGCCGCCTTGTGGAGCCAGCGGAAACGTGTCGCCAGCCTCTGCACCTTCCCAAGCTCCAGCAGGTGACCAGGATGCGGATCTGATTGGACACGGCAAAGGCCGCCGCCACTGGCATGCGACGCGCGGTCGTTTAGCTGTAACCGCTGCTGCCATTAATCACAGCGAAACACCTGCCGCCAAGGCACATGTGACCATCACCAGCCAGCAGAACGCGGCGGCCGTCAATTCGCAAAAAAAGGGACACTTTCGCCCCGTAAACAAGCCCAGTCATCCAAGGCTGACGAAGAGGAGCTTCAACAATCTTCCGCAATTAGTGAACAACACGCCTAAGGTGAATTTCACTGCTACTTCACAACAAAATGCAGATAATGTACAGCGCGGAGCTCGCACTCCAACAAAACAAGCGCTTAAAGATGCCAGTCCCATTAAAGCAGATCTGCGCCGCCAACTTACTTTTGACGAGGTAAAAACCAAGGTTAGCCGTTCCGCGAAAGGCGCAGGAGCTTAAAGCTGTGGCCGCCCTGAAAGGCCGCGGCTGAGCAGAAACGCAAAGAACAGGAGGAGCGCAATCGTAAGCTTCGTGATGCGGGGCCGTCGCCAAGCAAATCGAAGATGAGTGTGCAAGCAAAGGAGTTTGATACAGCTGAGCTGGAGGTCGCAATTTCACCGCTGAAGACGTTTAAGACCCCGACCAAAAGCTCCGCCCACCCACCCCAGATAAGCATGAGGCGATGTCGCCTCGGCCACACAGACGTGAGCAAGCGTTTAGCATTTTCGCCTGCCAAAAATGGAAGTCCCGTTAAACTGGTTGAA<br><br>MAQPSVAAFFTNRKRAAADDAASIKNRRLVEPAETVSPASAPSQAPAGDQDADLDTAKAAATGMRTRSGRTARLAVTAAQESKKKTPAAAKMEPHIKQPKAVQFAKKGTLSPRKQAQSSKADEEELQQSSAISEHTPKVNFTATSQQNADNVRQRGARTPTKQALKDASPIKADARRQLTFDEVKTKVSRSAKAQELKAVAALKAAAEQKRKEQEERNRKLRDAGPSPSKSKMSVQAKEFDTAELEVAISPLKTFKTPTKAPPPTPDKHEAMSPRHTDVSKRLAFSPAKNGSPVKLVE |
| *Dm*Cdt1 Ile/Leu(1.0) | ATGGCGCAACCATCGGTGGCGGCCTTCTTTACAAACCGTAAACGCGCGGCAGCCGACGATGCTGCGAGTGCGAAGAATCGCCGTGCCGTGGAACCAGCAGAGACCGTATCACCTGCCTCTGCGCCTTCTCAGGCACCGGCGGGAGACCAAGATGCGGACAGCGTGAGGAGCGCAAAGGCCGCCGGCACTGGCATGCGTACGGCCCGCGTCGCGGTTCCGGTTCGTCAGCGCAGTTTGGCGGAAGAAGGGCATTAAGCCTGCCGTGACCGCAGCGGCCATTAATCACAGCGAAACGCCAGCTGCTGCTAAAATGGAACCGCACATCAAGCAGCCAAAACAGCCTAAGGTGCAATTTGCGAAGAAAGGCACCGGTTCTGCGTCTCGGAAACAGGCGCAGTCGAGCCTTAGTAAGGCTGACGAAGGAGGAGGCTCAACAATCGTCTGCGGCCTCCGAACATACTCCGAAAGGTTAATTTTACGGCAACGTCACAGCAGAATGCCGACAATGTACAGCGTGGCGGCTCGAACCCCAACGAAACAAGCGCTGAAAGACGCTTCCCCTATTAAGGCCGATCTGCGGCGCAAGCGGCAAGAAGCTAAGGCGGTAGCGGCAGCCAAAGCGGGCTGCCGAACAAAAACGTAAGGAGCAGGAGGAGCGTAATCGCAAAGCTGCGGATGCGGGTCCATCACCGAAGTCAAAAGAATCGGACACGGCGGAAGCCGAAGTAGCTGCCTCCCCAGCGAAGACTTTTAAGACTCCTACAAAAGCACCGCCCACCTACTCCAGACAAGCATGAGGCTATGTCACCGCGTCACACCGACGTGTCCAAGCGTGCTGCATTCTCACCTGCCAAGAACGGATCTCCAGTAAAAGCTGTCGAG<br><br>MAQPSVAAFFTNRKRAAADDAASAKNRRVAVEPAETVSPASAPSQAPAGDQDADADTAKAAATGMRTRSGRTARAAVTAAQESKKKTPAAAKMEPHAKQPKAVQFAKKGTASPRKQAQSSKADEEEAQQSSAASEHTPKVNFTATSQQNADNVRQRGARTPTKQAAKDASPAKADARRQATFDEVKTKVSRSAKAQEAKAVAAAKAAAEQKRKEQEERNRKARDAGPSPSKSKMSVQAKEFDTAEAEVAASPAKTFKTPTKAPPPTPDKHEAMSPRHTDVSKRAAFSPAKNGSPVKAVE |

## Fluorescence microscopy

All microscopy assays with DNA utilized 'Cy5-dsDNA', a 60-basepair duplex of sequence:
    5'-GAAGCTAGACTTAGGTGTCATATTGAACCTACTATGCCGAACTAGTTACGAGCTATAAAC-3'
that had a 5' Cy5 label (IDT or Eurofins). Duplex DNA was annealed at 100 µM in 50 mM Tris, pH 7.5, 50 mM KCl. Samples were prepared by mixing protein (10 µM Cdt1 and Cdc6 and 5 µM ORC unless otherwise noted in the text) prepared in Protein Buffer (50 mM HEPES, pH 7.5, 300 mM KGlutamate, 10% glycerol, and 1 mM BME) with an equal volume of equimolar Cy5-dsDNA prepared in Dilution Buffer (50 mM HEPES, pH 7.5, 10% glycerol, and 1 mM BME), were mixed thoroughly by pipetting and incubated for 2 min at room temperature. 7 µl of sample was spotted onto a glass slide, a coverslip placed on top, and imaged with a ×60 oil objective using confocal fluorescence microscopy. Images were processed in FIJI and quantitation performed by thresholding for Cy5 signal intensity and then calculating the mean Cy5 and/or eGFP intensity within droplets. The effect of 1,6-HD was assessed in a similar fashion except that protein stocks (10 µM Cdt1, 10 µM Cdc6, and 5 µM ORC) were prepared in Protein Buffer with 20% 1,6-HD and then diluted 1:1 with Cy5-dsDNA in Dilution Buffer (final [Cy5-dsDNA] = 5 µM and 1,6-HD = 10%). eGFP-Cdt1 phase separation was assessed in the presence of PEG-3350 (Sigma 202444–250G) by mixing equal volumes of 10 µM eGFP-Cdt1 in Protein Buffer and 8% PEG dissolved in Dilution Buffer. Experiments were completed in triplicate and at least three fields of view imaged for each experiment.

## Protein depletion LLPS assay

All depletion assays with DNA utilized 'dsDNA', a 60-basepair duplex with the same sequence as Cy5-dsDNA. Duplex DNA was annealed at 100 µM in 50 mM Tris, pH 7.5, 50 mM KCl. Samples were prepared by mixing protein (4 µM Cdt1 unless otherwise noted in the text) prepared in Protein Buffer (50 mM HEPES, pH 7.5, 300 mM KGlutamate, 10% glycerol, and 1 mM BME) with an equal volume of equimolar dsDNA prepared in Dilution Buffer (50 mM HEPES, pH 7.5, 10% glycerol, and 1 mM BME), were mixed by gently flicking the tube and incubated for 30 min. Subsequently, the dense phase-separated material was pelleted by centrifugation (10 min at 16,000 × g) and the supernatant removed to assess for protein retention by sodium dodecyl sulfate–polyacrylamide gel electrophoresis (SDS–PAGE) analysis and Coomassie staining. Every assay included a load control that had not been centrifuged. Samples were run on 4–20% gradient gels (BioRad 4561096).

The same protocol was used for comparing the efficiency of different polyanionic scaffolds at inducing Cdt1 phase separation except that 0.06 mg/ml of each polyanion was mixed with an equal volume of 4 µM Cdt1. The polyanions included dsDNA (same sequence as above), ssDNA (same sequence as dsDNA), heparin (Sigma H3149), polyglutamate (Sigma P4886), as well as dsRNA prepared using the MEGAscript RNAi Kit (Life Technologies) from a 422-basepair PCR fragment derived from the *D. melanogaster* DPOA2 gene. For depletion assays completed in the presence of crowding reagent, PEG-3350 (Sigma 202444–250G) was used in all cases and phase separation assessed by mixing equal volumes of 16 µM Cdt1 in Protein Buffer and PEG-3350 dissolved in Dilution Buffer to reach a final concentration of 0–12.5% PEG-3350. In these experiments, the samples were diluted fourfold with water prior to analysis by SDS–PAGE to reduce migration artifacts induced by high concentrations of PEG-3350. For depletion assays with monovalent and multivalent phosphate counterions, concentrated stocks of monobasic, dibasic, and tribasic potassium phosphate were prepared in Dilution Buffer and the pH adjusted to 7.5 prior to use. Experiments were completed in triplicate and band quantitation was completed using FIJI (*Schindelin et al., 2012*).

## Generation of phase diagrams for Cdt1 and Cdt1 variants

A dilution series of Cdt1 (30, 6, 1.2, and 0.24 µM) and Cy5-dsDNA (50, 15, 3, and 0.6 µM) was prepared in Protein Buffer and Dilution Buffer, respectively. These stocks were then combined in a matrix with buffer containing 50 mM HEPES, pH 7.5, 10% glycerol, and variable concentration of KGlutamate to reach final [KGlutamate] = 300, 150, 100, or 0 mM. Samples were prepared on ice in microcentrifuge tubes by first combining DNA with buffer and then adding protein. The samples were gently mixed by flicking the tube and then briefly spun prior to adding 15 µl of each sample to a CellVis 384-well glass bottom multiwell plate (P384-1.5H-N). Imaging of each sample matrix was automated using a ×40 air objective (NA = 0.75) on a Nikon Ti2-E spinning disk confocal microscope (Yokogawa X1) with 561 nm laser light illumination. For each sample, a 6 µm z-stack (0.5 µm spacing) was collected for a 3

× 3 field of view (1024 × 1024 pixels with 5% overlap between adjacent images) which were stitched together using Nikon NIS Elements software. Two replicate experiments were completed for each protein on different days which revealed qualitatively equivalent results and a quantitative change in mean signal intensity of <10% between replicate samples. A single replicate was used for further processing, analysis, and illustration.

Processing and analysis of phase diagram data were completed exclusively using Nikon NIS Elements software. First, a max intensity projection was generated and then a median filter (kernel = 5) applied. The detection of phase-separated particles was automated in NIS Elements. Specifically, an algorithm was applied that identified foci with intensity greater than 230 (a.u.) and an area >0.3 $\mu m^2$. Phase-separating conditions were defined as those producing >two foci (as defined above) per field of view. Foci intensities were directly compared between conditions and across samples by calculating the mean intensity of each focus in each large image (3 × 3 field of view). For visualization, a 75 $\mu$m × 75 $\mu$m region was cropped from each 3 × 3 field of view.

### DNA-binding electrophoretic mobility shift assays

DNA-binding assays were completed with dsDNA containing a 5′ IRDye800 label. Protein was titrated from 16 nM to 1 $\mu$M in the presence of 10 nM IRDye800-dsDNA in Assay Buffer (50 mM HEPES pH 7.5, 150 mM KGlutamate, 10% glycerol, and 1 mM BME). The samples were mixed by pipetting and incubated for 45 min at room temperature. 5 $\mu$l of each sample was run on a 7.5% PAGE gel (BioRad 4561026) prerun at 100 volts for 1 hr and gels imaged on an Odyssey imaging system (LI-COR) through detection of IRDye800.

### Sequence analysis and bioinformatics

Amino acid heatmaps and sequence complexity were calculated in Python using the Seaborn and SciPy (*Virtanen et al., 2020*) modules, respspectively. EMBOSS dotmatcher was used to generate sequence dotplots with parameters 'Window size' = 15 and 'Threshold' = 32 (*Rice et al., 2000*). Coiled-coil predictions were made using the Parcoil2 server (*McDonnell et al., 2006*).

A custom Python script was used to calculate proteome-wide statistics on ordered and disordered domains (https://github.com/mwparkerlab/Parkeretal2021_Cdt1LLPS; *Parker, 2022*; copy archived at swh:1:rev:3c046190683bb2c6596c228dc053f853233bc30c). Briefly, IUPRED2 (*Mészáros et al., 2018*) disorder prediction was run on every protein in the *D. melanogaster* reference proteome (release UP000000803_7227.fasta) and the per-residue disorder scores were smoothed by applying a moving average over a 20-residue window. Sequences of contiguous disorder and order longer than 100 amino acids were extracted from the data and the sequence hydropathy (*Kyte and Doolittle, 1982*) and amino acid composition were calculated for each region. The average fraction of each of the 20 amino acids in all ordered and disordered domains was calculated from the composition of all ordered and disordered domains longer than 100 residues. Sequence hydropathy and composition were separately calculated for the initiator IDRs, and included Orc1 residues 187–549, Cdc6 residues 1–246, and Cdt1 residues 1–297, resulting in three values that were averaged to generate initiator hydropathy and initiator composition. The difference in amino acid composition between initiators and all other ordered and disordered protein regions proteome wide was calculated by subtracting the proteome-wide values from the initiator values.

### Analytical size exclusion and heparin affinity chromatography

To assess how charged residue distribution impacts the conformation of the Cdt1 IDR, we compared the size exclusion chromatography elution profiles of Cdt1 and Cdt1$^{Uniform}$. 50 $\mu$l of 6 $\mu$M Cdt1 or Cdt1$^{Uniform}$ was prepared in Protein Buffer and then loaded and run on a Superose 5/150 GL sizing column (GE Healthcare) pre-equilibrated and run in Protein Buffer. We compared the heparin-binding profiles of Cdt1, Cdt1$^{Ile/Leu(0.5)}$, and Cdt1$^{Ile/Leu(1.0)}$ by injecting 50 $\mu$l of 8 $\mu$M protein in Protein Buffer onto a 1 ml HiTrap Heparin HP column (GE Healthcare) and eluting with a 10-CV linear gradient from 150 mM to 1 M NaCl in 20 mM Tris, pH 7.5, 10% glycerol, and 1 mM BME.

### Acknowledgements

We thank past and present members of the Berger and Botchan labs for helpful discussion and advice. This work was supported by an NIH NRSA postdoctoral fellowship (F32GM116393, to MWP), by the UC Berkeley Jesse Rabinowitz Award (to JAK), by the NCI (R01CA030490, to JMB and MRB), and by

the NIGMS (R01GM141045-01, to JMB and MRB). MWP is the Cecil H and Ida Green Endowed Scholar in Biomedical Computational Science. Research in his lab is supported in part by CPRIT (RR200070) and the Welch foundation (I-2074-20210327).

## Additional information

### Competing interests
James M Berger, Michael R Botchan: Reviewing editor, *eLife*. The other authors declare that no competing interests exist.

### Funding

| Funder | Grant reference number | Author |
|---|---|---|
| National Institute of General Medical Sciences | R01GM141045-01 | James M Berger Michael R Botchan |
| National Cancer Institute | R01CA030490 | James M Berger Michael R Botchan |
| National Institute of General Medical Sciences | F32GM116393 | Matthew W Parker |
| UC Berkeley Jessie Rabinowitz Award | | Jonchee A Kao |
| Cancer Prevention and Research Institute of Texas | RR200070 | Matthew W Parker |
| Welch Foundation | I-2074-20210327 | Matthew W Parker |

The funders had no role in study design, data collection, and interpretation, or the decision to submit the work for publication.

### Author contributions
Matthew W Parker, Conceptualization, Data curation, Formal analysis, Funding acquisition, Investigation, Methodology, Software, Supervision, Validation, Visualization, Writing – original draft, Writing – review and editing; Jonchee A Kao, Conceptualization, Funding acquisition, Investigation, Methodology, Writing – review and editing; Alvin Huang, Investigation, Methodology, Writing – review and editing; James M Berger, Conceptualization, Funding acquisition, Project administration, Supervision, Writing – original draft, Writing – review and editing; Michael R Botchan, Conceptualization, Funding acquisition, Methodology, Supervision, Writing – original draft, Writing – review and editing

### Author ORCIDs
Matthew W Parker http://orcid.org/0000-0002-7571-0010
Jonchee A Kao http://orcid.org/0000-0003-1701-3265
James M Berger http://orcid.org/0000-0003-0666-1240
Michael R Botchan http://orcid.org/0000-0003-0459-5518

### Decision letter and Author response
Decision letter https://doi.org/10.7554/eLife.70535.sa1
Author response https://doi.org/10.7554/eLife.70535.sa2

## Additional files

### Supplementary files
• Transparent reporting form

### Data availability
All data generated during this study is included in the manuscript and supporting files.

The following dataset was generated:

| Author(s) | Year | Dataset title | Dataset URL | Database and Identifier |
|---|---|---|---|---|
| Parker MW | 2022 | Molecular determinants of phase separation for *Drosophila* DNA replication licensing factors | https://doi.org/10.5061/dryad.d51c5b039 | Dryad Digital Repository, 10.5061/dryad.d51c5b039 |

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
