## [Editor Report]

This paper studies the role of phase separation in replication initiation, with a focus on Cdt1. Sorting out the relative roles of phase separation and other mechanisms will require a detailed dissection of the amino acids driving phase separation, which can then be used to probe the role of phase separation in cells. Here, the authors perform extensive and comprehensive analyses of the amino-acid sequence requirements for Cdt1 phase separation finding important roles for charged and hydrophobic amino acids in mediating different aspects of the DNA-dependent phase separation observed.

---

## [Decision Letter]

**Decision letter after peer review:**

Thank you for submitting your article "Molecular determinants of phase separation for *Drosophila* DNA replication licensing factors" for consideration by *eLife*. Your article has been reviewed by 3 peer reviewers, including Stephen P Bell as the Reviewing Editor and Reviewer #3, and the evaluation has been overseen by Jessica Tyler as the Senior Editor.

The reviewers were impressed with the thorough analysis of the sequence determinants of Cdt1 LLPS. The studies clearly establish a role for interaction between basic amino acids and DNA and between hydrophobic residues (but not aromatic) amino acids as important drivers of Cdt1 LLPS. Although there is not a direct experiment showing the same is true for the other helicase loading proteins known to phase separate with Cdt1 (ORC and Cdc6), the authors show that the sequence makeup of the IDRs found in these proteins and likely to drive their LLPS are more similar to Cdt1 than other LLPS proteins. In addition to identifying a distinct type of LLPS-inducing sequence, these studies set the stage for critical experiments addressing the in vivo importance of these interactions for DNA replication initiation.

Essential revisions:

1) Phase diagrams should be generated and shown for wild-type Cdt1 and key mutants upon which the major conclusions are based (i.e. Cdt1-uniform, Cdt1-scrIDR, Cdt1-Ile/Leu(0.5), and Cdt1-Ile/Leu(1.0)

2) Revise the manuscript to acknowledge that other proteins that undergo LLPS do not require aromatic residues.

*Reviewer #1 (Recommendations for the authors):*

1) Figure 1. The font size in Figure 1 panels F and G is too small to read the superscripts. This occurs in some other Figures as well. Would it be possible to increase the font size?

2) Figure 1. The authors might consider to do a positive control for the hexane diol experiments of Figure 1B, C. I've noticed that in the authors earlier work they had purified FUS protein, which is listed in Table I as a hexandiol solubilized LLPS. If the authors have the hexandiol data for the FUS protein, it might be good to include it into Figure 1 as a positive control.

3) Lines 199-202: -K glutamate was used as the salt. While this is known to be the physiological salt for *E. coli*, would the authors know if this is the physiological salt for D.m.?

4) Lines 220-230: Were the mono, di and tri basic potassium phosphate all at the same pH? If not, it would be good to state the pH.

5) Figure 3, lines 307. The 50% reduction in forming the LLPS seems quite substantial for deletion of residues 201-297. It suggests this region has the most tendency toward forming the LLPS. I wonder if one were to make a deletion of 1-200, and to use a 2X 201-297, if the LLPS would be efficient. Just a suggestion – not asking to do it.

6) In the Discussion, it would be good to mention just how conserved – or not – the IDR of Ctd1 is among metazoans. Is it generally well preserved sequence? In which case one might think it is for kinases, in light of this report? Regardless – it would be good to give readers a better idea of the sequence conservation of the IDR in metazoans.

7) The Results section says "Results and Discussion" – yet there is a separate "Discussion". Is this OK in *eLife*?

8) Discussion, and elsewhere. The authors discuss the IDR and LLPS demonstrated here as an initiator specific type of LLPS. But is it possible that it is more general – in being a "DNA specific" LLPS, and not just initiation. If the authors do not feel that point could increase the audience/interest in their work, they are welcome to omit this comment.

*Reviewer #2 (Recommendations for the authors):*

I think this paper would benefice from the authors providing an in-depth discussion of their ideas, and more importantly, the limitations of this study. Although it can be technically challenging, neither this, nor their previous manuscript address phase separation in vivo and to this point it is still unclear whether phase separation plays a role for replication initiation.

*Reviewer #3 (Recommendations for the authors):*

The addition of a test of the generality of the authors conclusions by creating a couple of IDR mutants (e.g. one eliminating branched chain hydrophobic residues and one eliminating aromatic residues) would be an important test of the authors' hypothesis.

---

## [Author Response]

Essential revisions:1) Phase diagrams should be generated and shown for wild-type Cdt1 and key mutants upon which the major conclusions are based (i.e. Cdt1-uniform, Cdt1-scrIDR, Cdt1-Ile/Leu(0.5), and Cdt1-Ile/Leu(1.0)

We thank the reviewers for this excellent suggestion. We have now generated phase diagrams for Cdt1, Cdt1^Uniform^, Cdt1^ScrIDR^, Cdt1^Ile/Leu(0.5)^ and Cdt1^Ile/Leu(1.0)^ under variable protein and potassium glutamate concentrations. These data provide strong additional support for the major conclusions of our paper by providing a quantitative comparison of the DNA-dependent phase separation propensity of each variant. Specifically, they demonstrate a 25-fold reduction in the critical concentration for both Ile/Leu variants, as well as a significant reduction in the ability of these variants to concentrate nucleic acid within the condensed phase that forms at high protein concentrations. In addition to this, the analysis revealed unappreciated differences between other Cdt1 variants. For example, at protein concentrations exceeding the critical concentration, the uniform variant is more sensitive to salt than either wild-type Cdt1 or the scrambled variant, and the scrambled variant is less sensitive to salt at concentrations around the critical concentration. These phase diagrams have now been added to main text figures and are discussed.

2) Revise the manuscript to acknowledge that other proteins that undergo LLPS do not require aromatic residues.

We agree with the reviewers that there likely exist numerous phase-separating sequences, both discovered and undiscovered, that do not require aromatic residues for LLPS. We have therefore fully reworked our introductory paragraph on molecular mechanisms of LLPS to clearly express this viewpoint and discuss alternative mechanisms of LLPS, such as those systems thought to be driven purely by electrostatic interactions. We now state:

“Literature reports of phase-separating sequences are increasing rapidly, yet relatively few of these studies define the molecular basis for LLPS. […] An understanding of the molecular rules that govern IDR sorting mechanisms is still in its infancy.”

Reviewer #1 (Recommendations for the authors):1) Figure 1. The font size in Figure 1 panels F and G is too small to read the superscripts. This occurs in some other Figures as well. Would it be possible to increase the font size?

We have now increased the font size of all figure text that contains superscripts.

2) Figure 1. The authors might consider to do a positive control for the hexane diol experiments of Figure 1B, C. I've noticed that in the authors earlier work they had purified FUS protein, which is listed in Table I as a hexandiol solubilized LLPS. If the authors have the hexandiol data for the FUS protein, it might be good to include it into Figure 1 as a positive control.

We agree with the reviewer that this would be a good positive control. However, the eGFP-FUS stock we generated for our previous assays, which were completed below the FUS critical concentration, is not at a sufficiently high concentration to assess FUS phase separation in isolation. We apologize that we will not be able to include the suggested experiment.

3) Lines 199-202: -K glutamate was used as the salt. While this is known to be the physiological salt for *E. coli*, would the authors know if this is the physiological salt for D.m.?

To our knowledge there are no reports of the intracellular ion and metabolite concentrations specifically in *Drosophila* cells. However, these numbers are available for yeast and human, as well as *E. coli*. The most abundant intracellular cation in mammalian cells, yeast and *E. coli* is potassium (approximately 100-150 mM for mammalian cells, 300 mM for yeast cells and 200 mM for *E. coli*). However, inorganic ions represent only a tiny fraction of the total cellular anion pool which is instead dominated primarily by organic phosphates and acids. Most abundant among these in all species currently analyzed is glutamate. Thus, as seen for *E. coli*, the most physiologically relevant salt is potassium glutamate. We point the reviewer to the following references:

Park, J. O., Rubin, S. A., Xu, Y. F., Amador-Noguez, D., Fan, J., Shlomi, T., and Rabinowitz, J. D. (2016). Metabolite concentrations, fluxes and free energies imply efficient enzyme usage. Nature Chemical Biology 2016 12:7, 12(7), 482–489. https://doi.org/10.1038/nchembio.2077

Andersen, O. S. (2013). Cellular Electrolyte Metabolism. In Encyclopedia of Metalloproteins (pp. 580–587). Springer New York.

4) Lines 220-230: Were the mono, di and tri basic potassium phosphate all at the same pH? If not, it would be good to state the pH.

Each of the phosphate buffer stocks was adjusted to pH=7.5. While we included this point in the methods section, we now also include this important technicality in the Results section. Lines 220-230 now state:

“Using the depletion assay we found that monobasic potassium phosphate (pH 7.5) was unable to induce phase separation of Cdt1 up to the highest concentration tested (625 mM). Conversely, we observed significant depletion of Cdt1 at the highest concentration of dibasic potassium phosphate (625 mM, pH 7.5), as well as for the three highest concentrations of tribasic potassium phosphate (400, 500 and 625 mM, pH 7.5).”

5) Figure 3, lines 307. The 50% reduction in forming the LLPS seems quite substantial for deletion of residues 201-297. It suggests this region has the most tendency toward forming the LLPS. I wonder if one were to make a deletion of 1-200, and to use a 2X 201-297, if the LLPS would be efficient. Just a suggestion – not asking to do it.

We thank the reviewer for this excellent suggestion. Our ongoing work is taking such a strategy, using the 201-297 region and, in particular, the highly hydrophobic region between residues 244-LEVLI-248, as a model sequence for de novo engineering of DNA-dependent phase separating peptides.

6) In the Discussion, it would be good to mention just how conserved – or not – the IDR of Ctd1 is among metazoans. Is it generally well preserved sequence? In which case one might think it is for kinases, in light of this report? Regardless – it would be good to give readers a better idea of the sequence conservation of the IDR in metazoans.

The question of conservation is interesting given that amino acid ordering is not required for Cdt1 phase separation (Figure 3), which may thus relieve evolutionary pressure on the specific ordering of amino acids. We have now assessed conservation for Cdt1 orthologs from both across the metazoan lineage, as well as for Drosophilidae Cdt1s, and we have calculated the number of CDK/Cyc motifs in each ortholog. These results led to two important insights: First, the Cdt1 IDR is not well conserved across metazoans but is highly conserved amongst Drosophilidaes. Second, despite the absence of sequence conservation across metazoans, the vast majority of metazoan Cdt1 orthologs contain CDK/Cyc consensus motifs (with the exception of hydra and rotifer Cdt1). We now include this as supplemental data with the following added discussion:

“Future work will need to address whether heteromeric inter-IDR interactions (e.g. Orc1^IDR^-Cdc^IDR^) are indeed governed strictly by sequence composition or whether linear sequence motifs facilitate specific interactions, as was recently suggested for human Orc1 and Cdc6 (Hossain et al., 2021). […] These results suggest that certain classes of disordered domains can have conserved functionality in the absence of linear sequence identity.”

And,

“Sequence information within the Orc1 and Cdt1 IDR’s is high, and while saltatory leaps seem to have occurred between phyla (Figure 5—figure supplement 2B), the conservation of sequence homology within the *Drosophila* genus (Figure 5—figure supplement 2C and (Parker et al., 2019)), over millions of years, is remarkable and indicates an evolutionary pressure on sequence order for unknown function. […] Notably, consensus motifs for CDK-dependent phosphorylation (full site = [S/T]PX[R/K] and minimal site = [S/T]P) are conserved in a majority of metazoan Cdt1s (Figure 5—figure supplement 2D) and in all sequenced Drosophilidae Cdt1 orthologs (Figure 5—figure supplement 2E), leading us to predict that phospho-tunable phase separation is a broadly conserved mechanism for regulating metazoan replication licensing.”

7) The Results section says "Results and Discussion" – yet there is a separate "Discussion". Is this OK in eLife?

We thank the reviewer for catching this typo. We now have clearly marked “Results” and “Discussion” sections.

8) Discussion, and elsewhere. The authors discuss the IDR and LLPS demonstrated here as an initiator specific type of LLPS. But is it possible that it is more general – in being a "DNA specific" LLPS, and not just initiation. If the authors do not feel that point could increase the audience/interest in their work, they are welcome to omit this comment.

This is an excellent point, and we agree. While we are calling this type of IDR an “initiator-type” for the factors in which it was identified, we recognize that sequences with similar composition may be present in other chromatin-binding proteins. As the reviewer suggests, this point may be of broad interest to the community, and we have now revised and expanded our discussion to communicate this. Specifically, we write:

“The identification of an IDR in Chiffon with compositional similarity to the initiator-type IDRs suggests that such sequences may be present in chromatin-associated factors beyond replication licensing components. […] Thus, our future work will aim to develop algorithms that accurately identify proteins proteome-wide that possess disordered domains with compositional homology to initiator IDRs and to understand how these sequences impact protein functional dynamics.”

Reviewer #2 (Recommendations for the authors):I think this paper would benefice from the authors providing an in-depth discussion of their ideas, and more importantly, the limitations of this study. Although it can be technically challenging, neither this, nor their previous manuscript address phase separation in vivo and to this point it is still unclear whether phase separation plays a role for replication initiation.

We agree with the reviewer that a discussion of the limitations of our study is warranted, particularly since we have not yet explicitly demonstrated the relevance of phase separation to the replication licensing reaction. We have modified our discussion to state:

“Defining the molecular ‘grammar’ that encodes LLPS in metazoan initiators likewise has implications for understanding the impact of initiator phase separation in vivo. […] Thus, we predict that initiators condense on the surface of mitotic chromosomes without altering their shape.”

Reviewer #3 (Recommendations for the authors):The addition of a test of the generality of the authors conclusions by creating a couple of IDR mutants (e.g. one eliminating branched chain hydrophobic residues and one eliminating aromatic residues) would be an important test of the authors' hypothesis.

We agree that establishing the generality of our conclusions is an important future step. Towards this end, we attempted to express and purify the wild-type Orc1 IDR and an Orc1 mutant with all Leu and Ile residues mutated to Ala for phase separation assays. We were able to dramatically improve expression of the wild-type Orc1 sequence compared to our previous attempts (Parker et al., *eLife*, 2019) through generation of a codon optimized construct and fusion to MBP (the MBP tag was removed prior to the final size exclusion polishing step). This resulted in > 2 mg of pure protein per L of culture. Unfortunately, even with codon optimization and an MBP tag we could not produce sufficient quantities of the Orc1^Ile/Leu(1.0)^ mutant for biochemical assays. In Author response image 1 we show the elution fractions from the heparin purification step of both the wild-type Orc1 (left) and mutant (right) sequence. While the elution profile looks similar between the WT and mutant sequence, indicating that the mutant Orc1 IDR still possesses heparin affinity (same as Cdt1^Ile/Leu(1.0)^), there was insufficient mutant protein to proceed with the final purification steps (TEV cleavage of the MBP tag, ortho nickel purification and a final size exclusion chromatography run).

**Author response image 1. sa2fig1:** 

Our inability to produce a suitable Orc1 mutant prompted us to consider alternative biochemical experiments that may support the generality of our claims to the other initiation factors. Our data suggest a mechanism of Cdt1 LLPS that relies on both heteromeric electrostatic Cdt1-DNA interactions and homomeric hydrophobic inter-IDR interactions. We therefore used the pelleting assay and purified ORC holocomplex and Cdc6 to test three assumptions of this model:

(1) The salt-sensitivity of *DNA-dependent* ORC and Cdc6 phase separation. This would be suggestive that heteromeric electrostatic initiator-DNA interactions are important for LLPS.

(2) *DNA-independent* phase separation of ORC and Cdc6 in the presence of crowding reagent. This would be suggestive of direct homomeric inter-IDR interactions.

(3) The salt-insensitivity of crowding reagent-induced (*DNA-independent*) ORC and Cdc6 phase separation. This would be suggestive of non-electrostatic inter-IDR interactions which, we argue, would support the role of hydrophobics since aromatics are largely excluded from the initiator IDR sequences.

Consistent with a shared mechanism of LLPS for ORC, Cdc6 and Cdt1, we confirm the above three claims, demonstrating that ORC and Cdc6 can be induced to phase separate either in the presence of DNA or PEG, but that DNA-induced LLPS is salt-sensitive while PEG-induced phase separation is not. We believe these data provide strong support for our prediction that initiators share a common LLPS mechanism. These data are now included as a supplemental figure and discussed in text.